# Enhancing radiation-resistance and peroxidase-like activity of single-atom copper nanozyme via local coordination manipulation

Jiabin Wu [1,11], Xianyu Zhu[2,3,11], Qun Li[4], Qiang Fu [5,6] ✉, Bingxue Wang[5], Beibei Li[1], Shanshan Wang[7], Qingchao Chang[2], Huandong Xiang[2,8], Chengliang Ye[1], Qiqiang Li[3], Liang Huang [4], Yan Liang [2] ✉, Dingsheng Wang [1], Yuliang Zhao [2,8] & Yadong Li [1,9,10] ✉

The inactivation of natural enzymes by radiation poses a great challenge to their applications for radiotherapy. Single-atom nanozymes (SAzymes) with high structural stability under such extreme conditions become a promising candidate for replacing natural enzymes to shrink tumors. Here, we report a $CuN_3$-centered SAzyme ($CuN_3$-SAzyme) that exhibits higher peroxidase-like catalytic activity than a $CuN_4$-centered counterpart, by locally regulating the coordination environment of single copper sites. Density functional theory calculations reveal that the $CuN_3$ active moiety confers optimal $H_2O_2$ adsorption and dissociation properties, thus contributing to high enzymatic activity of $CuN_3$-SAzyme. The introduction of X-ray can improve the kinetics of the decomposition of $H_2O_2$ by $CuN_3$-SAzyme. Moreover, $CuN_3$-SAzyme is very stable after a total radiation dose of 500 Gy, without significant changes in its geometrical structure or coordination environment, and simultaneously still retains comparable peroxidase-like activity relative to natural enzymes. Finally, this developed $CuN_3$-SAzyme with remarkable radioresistance can be used as an external field-improved therapeutics for enhancing radio-enzymatic therapy in vitro and in vivo. Overall, this study provides a paradigm for developing SAzymes with improved enzymatic activity through local coordination manipulation and high radioresistance over natural enzymes, for example, as sensitizers for cancer therapy.

Generally, natural enzymes exposed to radiation such as X- and γ-rays can suffer a permanent alteration in their structures, resulting in their inactivation to some extent through both direct and indirect actions[1–5]. Because water constitutes nearly 80% of the composition of most biological tissues, X-ray can ionize water molecules to break the hydrogen-oxygen bonds, generating a series of chemically reactive species (e.g., $H_2O_2$, ·OH, ·H, and $e_{aq}^-$)[6–8]. Sequentially, these species diffuse towards the surface of natural enzymes to damage their biochemical structures by chemical reactions. Meanwhile, X-ray can also interact with the atoms of natural enzymes, leading to the chain of physical and chemical events that eventually cause their structural changes and even chemical modifications[9–12]. Therefore, the inactivation of natural enzymes by X- and γ-rays dramatically hinders their various applications[13]. For example, catalase, which can reverse the

tumor hypoxia microenvironment via decomposing $H_2O_2$ into $O_2$, would lose its enzymatic activity during the course of radiotherapy[14], consequently decreasing radiotherapeutic efficacy[15,16]. Therefore, a major central issue of current research is to develop artificial enzymes with excellent radiation-resistance as well as superior enzymatic activity, aiming at addressing the limitations of natural enzymes to fulfilment of their actual application in radiotherapy.

Nanoenzymes, which exhibit intrinsic enzyme-mimetic activity, combine the advantages of biocatalysts and nanomaterials to confer unique capabilities beyond natural enzymes, including low cost, high stability, and easy storage[17–20]. Owing to artificial enzyme engineering capable of manipulating biocatalytic active sites at both atomic and molecular levels, nanoenzymes with peroxidase-, superoxide dismutase-, and catalase-like enzymatic activities are continually designed through the inspiration from the structure of natural enzymes[21–26]. So far, significant effort has been focused on understanding those nanozymes for tumor diagnosis and therapy from in vitro to in vivo[27–32]. To further enhance the enzymatic activity, single-atom nanoenzymes (SAzymes) with designable geometric structures and optimized atom utilization have been studied for enzymatic therapy via generating therapeutic species from intracellular substances[18,33–40]. In particular, Cu SAzymes with densely isolated $CuN_4$ sites ($CuN_4$-SAzyme) have been demonstrated to exhibit higher peroxidase-like activity than copper oxide ($CuO_x$) nanoparticles[41–45]. However, this Cu-$N_4$ coordination structure still suffers from weak adsorption of $H_2O_2$, leading to the direct detachment of intermediate species. To overcome this issue, an alternative approach is to modulate the coordination number of single Cu sites, so as to promote the adsorption and dissociation of $H_2O_2$ at the active sites and boost the activity of Cu-based SAzymes towards peroxidase-like reactions[46–50]. On the other hand, carbon materials are highly resistant to radiation damage[51–55] and can be served as the substrate of SAzymes to endow them with high structural stability under X-ray and even high-energy γ-ray irradiation. This feature is able to stabilize the geometrical structure of active sites and maintain the enzyme activity against radiation. Hence, it is conceived that, by precisely regulating local coordination environment, such as coordination number, of single Cu sites supported on two-dimensional (2D) carbon nanostructures, $CuN_x$-SAzymes with excellent enzymatic activity and radioresistance will be realized to overcome the drawbacks associated with traditional nanoenzymes and natural enzymes, hopefully resulting in the concurrent higher therapeutic efficacy and fewer side effect to healthy tissues.

In this study, we fabricate a powerful single-atom Cu enzyme with Cu-$N_3$ coordination ($CuN_3$-SAzyme) via regulating local coordination number. The enzymatic activity and kinetics of this resulting $CuN_3$-SAzyme are much higher than those of $CuN_4$-SAzyme and $CuO_x$ nanozymes. Density functional theory (DFT) calculations demonstrate that the designed $CuN_3$ active moiety endows $CuN_3$-SAzyme with remarkable peroxidase-like activity. Furthermore, $CuN_3$-SAzyme is more stable than natural enzyme peroxidase when exposed to X-ray with the radiation dose up to 500 Gy. Interestingly, the introduction of X-ray can enhance the peroxidase-like activity and kinetics of $CuN_3$-SAzyme. Combining superior radioresistance and excellent enzymatic activity, $CuN_3$-SAzyme can be utilized for enhanced radio-enzymatic therapy to resolve the crucial issues facing the application of natural enzymes.

## Results

### Synthesis and characterization of $CuN_x$-SAzymes

The synthesis of $CuN_3$-SAzyme and $CuN_4$-SAzyme is schematically illustrated in Fig. 1a. Taking $CuN_3$-SAzyme as an example, KCl crystals as the template were mixed with a methanol solution containing $Cu(NO_3)_2$ and 2-methylimidazole to form Cu-MOFs layers and the as-obtained precursor was calcinated at 850 °C for 2 h. After the removal of KCl template and etching with $H_2SO_4$, 2D carbon nanosheets

supporting $CuN_3$ active moieties were obtained. Transmission electron microscopy (TEM) images demonstrate a typical 2D feature with atomically smooth surface for $CuN_3$-SAzyme (Fig. 1b), which is consistent with the result of Raman analysis (Supplementary Fig. 1), while a polyhedral morphology with a porous structure for $CuN_4$-SAzyme (Supplementary Fig. 2). Their X-ray diffraction (XRD) patterns show main broad diffraction peaks at ~26° corresponding to graphitic carbon but no significant characteristic peaks ascribed to Cu-based nanoparticles (Supplementary Fig. 3). Energy-dispersive spectroscopy (EDS) mapping analysis displays uniform dispersion of C, N, and Cu elements throughout the whole architectures of $CuN_3$-SAzyme and $CuN_4$-SAzyme, respectively (Fig. 1c, d and Supplementary Fig. 4). X-ray photoelectron spectroscopy (XPS) analysis was performed to determine the chemical states of C and N elements on $CuN_x$-SAzymes (Supplementary Figs. 5, 6). Typically, C 1s spectrum exhibits three peaks associated with C = C/C-C (~284.7 eV), C-N (~285.8 eV), and O = C-C (~288.5 eV) and N 1s spectrum can be deconvoluted into three peaks at binding energies of ~398.6 eV, ~400.1 eV, and ~401.3 eV, which are attributed to pyridinic N, pyrrolic N, and graphitic N, respectively. Inductively coupled plasma optical emission spectrometry (ICP-OES) was further applied to quantify the Cu content, which is 2.98 wt% for $CuN_3$-SAzyme and 1.21 wt% for $CuN_4$-SAzyme. In addition, the aberration-corrected high angle annular dark-field scanning transmission electron microscopy (AC HAADF-STEM) was performed to investigate $CuN_x$-SAzymes at the atomic scale (Fig. 1e–g and Supplementary Fig. 7).

### Comparison of atomic structure of $CuN_3$-SAzyme with $CuN_4$-SAzyme

The chemical state and coordination environment of both $CuN_3$-SAzyme and $CuN_4$-SAzyme were analyzed using Cu K-edge X-ray absorption fine structure (XAFS) spectroscopy. The X-ray absorption near-edge structure (XANES) spectrum, which is sensitive to the three-dimensional atomic arrangement around the center metal, serves as a powerful tool for identifying the atomic configurations. Figure 2a shows the Cu K-edge XANES spectra of $CuN_x$-SAzyme, $CuO_x$ nanozymes, and Cu foil. It is noteworthy that the absorption edges of $CuN_3$-SAzyme and $CuN_4$-SAzyme are located between those of $CuO_x$ nanozymes. For $CuN_3$-SAzyme, the absorption edge is closer to that of $Cu_2O$ nanozyme, indicating that the atomically dispersed Cu species have a positive charge with a chemical valence close to +1. This finding aligns with the Cu 2p XPS spectrum of $CuN_3$-SAzyme, which shows a peak at the binding energy of 931.4 eV (Fig. 2b). The presence of trace amount of $Cu^{2+}$ in $CuN_3$-SAzyme could be attributed to the acid etching process. In contrast, the absorption edge of $CuN_4$-SAzyme is much closer to that of CuO nanozyme and its Cu 2p XPS spectrum displays a main peak at a binding energy of 934.4 eV[56,57]. These observations suggest that the atomically dispersed Cu species in $CuN_4$-SAzyme have a chemical valence close to +2. A minor presence of $Cu^+$ (931.4 eV) is likely due to the reduction of $Cu^{2+}$ by ammonia generated during the pyrolysis process.

Figure 2c displays the Fourier transformed extended Cu K-edge X-ray absorption fine structure (FT-EXAFS) spectra of $CuN_x$-SAzymes, using $CuO_x$ nanozymes and Cu foil as references. Both $CuN_3$-SAzyme and $CuN_4$-SAzyme display a main peak around 1.5 Å, which corresponds to the Cu-N peak, with no obvious peak at ~2.3 Å, indicating the absence of Cu-Cu coordination. This confirms that no metallic crystalline Cu species are formed and that only isolated single Cu atoms are present, in agreement with the AC HAADF-STEM measurement results. To gain further structural insights, we performed least-squares fits to the $R$-space FT-EXAFS plots of $CuN_x$-SAzymes, $CuO_x$ nanozymes, and Cu foil (Fig. 2d and Supplementary Fig. 8). The $k$-space EXAFS fitting plots are shown in Fig. 2e and Supplementary Fig. 9, with refined parameters listed in Supplementary Table 1. The best-fitting results demonstrate that the dominant peak at ~1.5 Å is due to the Cu-N first

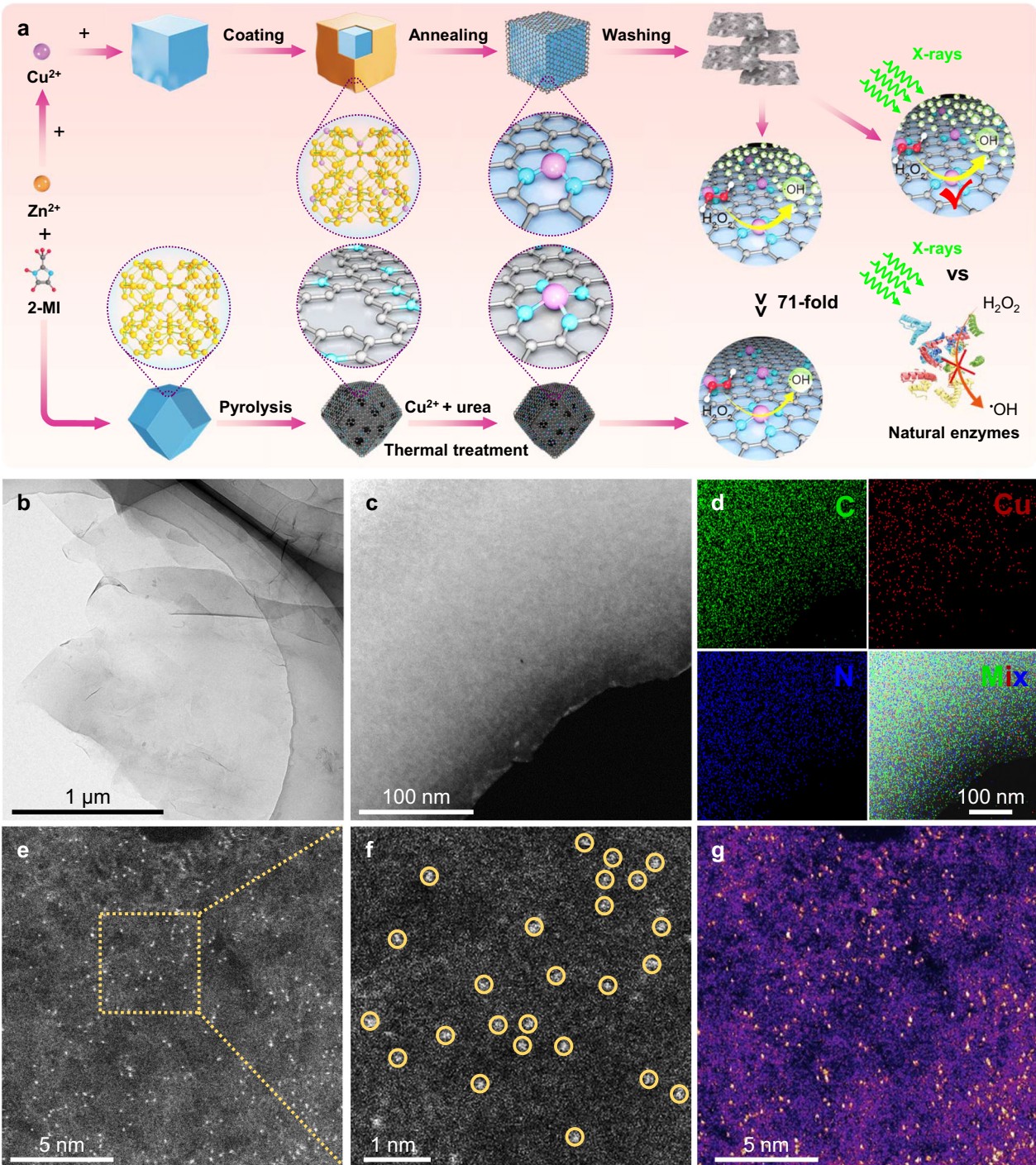

**Fig. 1 | Characterization of CuN₃-SAzyme. a** Schematic diagram of the synthesis strategy of CuN₃-SAzyme and CuN₄-SAzyme. 2-methylimidazole: 2-MI. **b, c** HR-TEM (**b**) and HAADF-STEM (**c**) images of CuN₃-SAzyme. **d** Corresponding EDS mapping of CuN₃-SAzyme. C, green; N, blue; Cu, red. **e** Atomic-level HAADF-STEM image of CuN₃-SAzyme. **f** Enlarged HAADF-STEM image of the marked area in (**e**). **g** Corresponding surface intensity map of **e**, the yellow dots are Cu atoms. Three times each morphology characterization was repeated independently with similar results (**b–g**). Representative images are presented.

shell coordination. Given that the coordination number of Cu is three, the proposed structural model of CuN₃ active moieties is illustrated in Fig. 2f. The $q$-space EXAFS plots of CuNₓ-SAzymes, CuOₓ nanozymes, and Cu foil are shown in Supplementary Fig. 10.

EXAFS wavelet transform (WT) analysis was employed to distinguish the backscattering atoms, even when there was significant overlap in $R$ space. As depicted in Fig. 2g, the WT contour plot of CuN₃-SAzyme exhibits a single intensity peak around 5 Å⁻¹, indicating the coordination between Cu and light atoms. This result is consistent with

the WT contour plot of CuOₓ nanozymes. Unlike Cu foil, the WT contour plot of CuN₃-SAzyme does not show an intensity peak at 7 Å⁻¹, which is associated with Cu-Cu interactions. This absence confirms that the Cu species in CuN₃-SAzyme are atomically dispersed without metal-derived crystalline structures (Fig. 2h, i and Supplementary Fig. 11).

**Evaluation of peroxidase-like activity of CuN₃-SAzyme**

Inspired by the chemical microenvironments capable of altering the activity of SAzymes, we next estimated and compared the peroxidase-

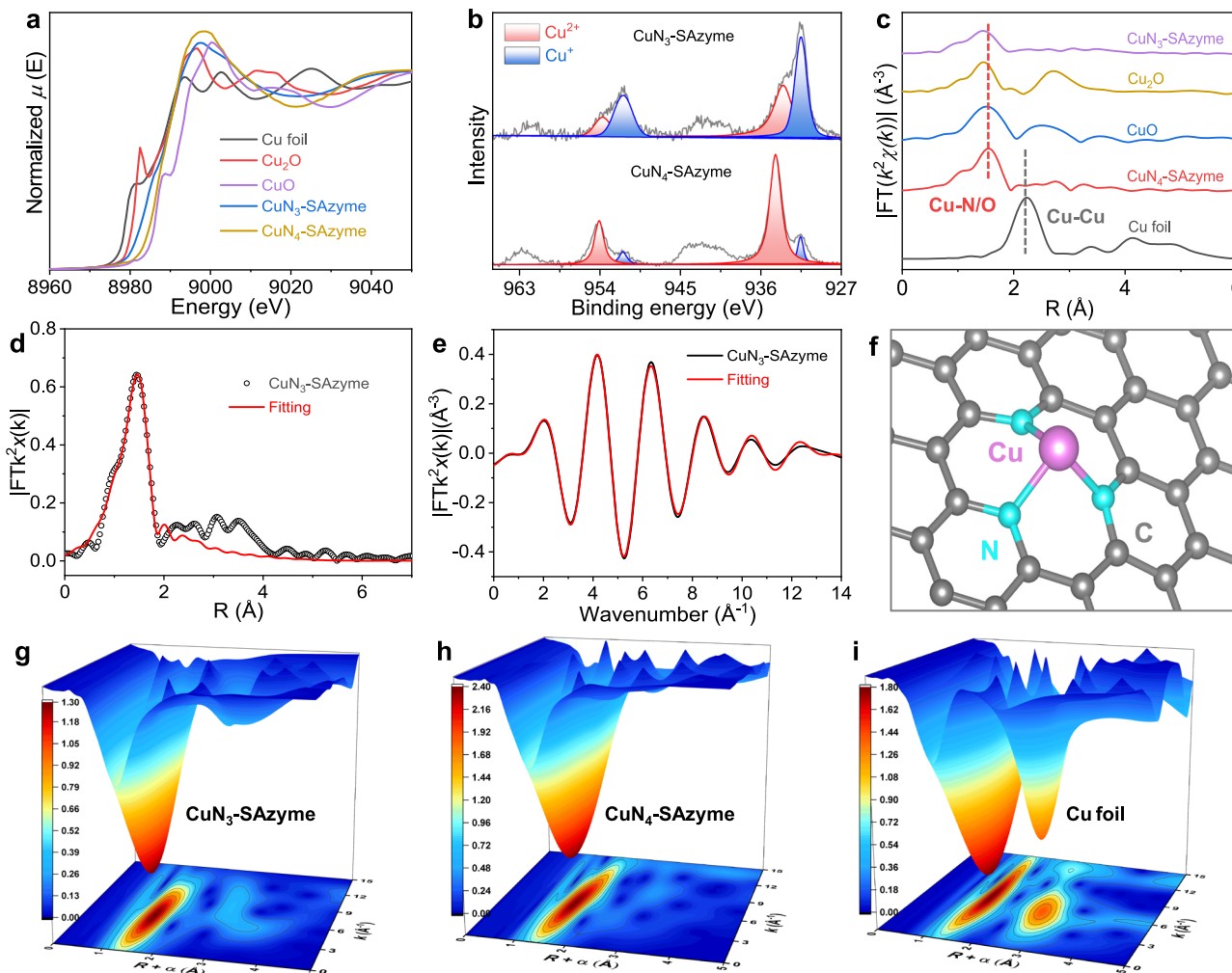

**Fig. 2 | Atomic structural analysis of CuN₃-SAzyme by XAFS. a** Cu K-edge XANES spectra of CuN₃-SAzyme and reference samples. **b** XPS spectra of Cu 2p orbital of CuN₃-SAzyme and CuN₄-SAzyme. **c** Fourier-transformed magnitudes of experimental Cu K-edge FT-EXAFS signals of CuN₃-SAzyme along with reference samples at *R* space. **d** *R*-space FT-EXAFS of CuN₃-SAzyme was fitted to obtain the quantitative local structure of Cu. **e** *q*-space FT-EXAFS fitting plots of CuN₃-SAzyme. **f** Schematic model of CuN₃-SAzyme. **g–i** WT-EXAFS curves of CuN₃-SAzyme (**g**), CuN₄-SAzyme (**h**), and Cu foil sample (**i**).

like activities of CuNₓ-SAzymes. Similar to natural horseradish peroxidase (HRP), both CuN₃-SAzyme and CuN₄-SAzyme show intrinsic peroxidase-like activity towards typical peroxidase substrates, such as 3,3′,5,5′-tetramethylbenzidine (TMB), 1,2-diaminobenzene (OPD), and 2,2′-azino-bis (3-ethylbenzothiazoline-6-sulfonic acid) (ABTS), to produce distinct color reactions in the presence of $H_2O_2$ (Fig. 3a). Taking TMB as a typical substrate, the absorbance of oxidized TMB increases immediately for CuN₃-SAzyme, which is much faster than those of CuN₄-SAzyme as well as CuOₓ nanozymes. This indicates that CuN₃-SAzyme displays the highest enzymatic activity among those tested Cu-based nanozymes. The calculated apparent specific activity of CuN₃-SAzyme in pH 3.54 (11.33 U mg⁻¹) is about 71-fold higher than that of CuN₄-SAzyme (0.16 U mg⁻¹), more than 12-fold higher than that of CuO nanozyme (0.91 U mg⁻¹), and almost 5-fold higher than that of Cu₂O nanozyme (2.08 U mg⁻¹) (Fig. 3b). Conversely, Cu-free N-doped carbon support shows no significant catalytic performance under the same conditions (Supplementary Fig. 12). These results indicate that regulating the coordination structure, in particular coordination number, of the Cu sites enables orders of magnitude enhancement of the peroxidase-like activity of Cu-based nanozymes. In addition, the catalase-like activity of CuN₃-SAzyme was also determined and calculated to be 0.40 U mg⁻¹ (Supplementary Fig. 13). This value is lower than the per-oxidase activity of CuN₃-SAzyme, showing its catalytic specificity.

These nanozymes were found to follow the Michaelis-Menten kinetics and the steady-state kinetic parameters were further determined to quantitatively compare their catalytic characteristics (Supplementary Fig. 14). Notably, at pH 3.54, CuN₃-SAzyme exhibits better catalytic efficiency ($k_{cat}/K_m = 6.56 \times 10^3$ M⁻¹ min⁻¹) and higher selectivity ($K_m = 1.61 \times 10^{-3}$ M) than those of CuN₄-SAzyme ($k_{cat}/K_m = 1.46 \times 10^2$ M⁻¹ min⁻¹, $K_m = 1.42 \times 10^{-2}$ M), as well as CuOₓ nanozymes ($k_{cat}/K_m = 19.30$ M⁻¹ min⁻¹, $K_m = 7.53 \times 10^{-3}$ M for Cu₂O nanozyme; $k_{cat}/K_m = 31.30$ M⁻¹ min⁻¹, $K_m = 2.11 \times 10^{-3}$ M for CuO nanozyme) (Supplementary Table 2). The kinetic parameters for $H_2O_2$ substrate were also measured to confirm excellent enzymatic activity of CuN₃-SAzyme relative to CuN₄-SAzyme (Supplementary Fig. 15). As expected, the $k_{cat}$ and $k_{cat}/V_{max}$ values of Cu₃-SAzyme on $H_2O_2$ substrate ($k_{cat} = 9.41$ min⁻¹, $k_{cat}/K_m = 1.43 \times 10^2$ M⁻¹ min⁻¹) is much higher compared to CuN₄-SAzyme ($k_{cat} = 1.27$ min⁻¹, $k_{cat}/K_m = 1.69$ M⁻¹ min⁻¹) and CuOₓ nanozymes ($k_{cat} = 5.88 \times 10^{-2}$ min⁻¹, $k_{cat}/K_m = 0.52$ M⁻¹ min⁻¹ for Cu₂O nanozyme; $k_{cat} = 4.27 \times 10^{-2}$ min⁻¹, $k_{cat}/K_m = 0.81$ M⁻¹ min⁻¹ for CuO nanozyme) at pH 3.54, indicative of higher enzymatic activity of Cu₃-SAzyme (Supplementary Table 3). The similar trend is also observed at pH 6.49 (Supplementary Fig. 16). These results facilitate the establishment of a relationship between coordination environments and catalytic performance for rational design of SAzymes with excellent enzymatic activity beyond natural enzymes.

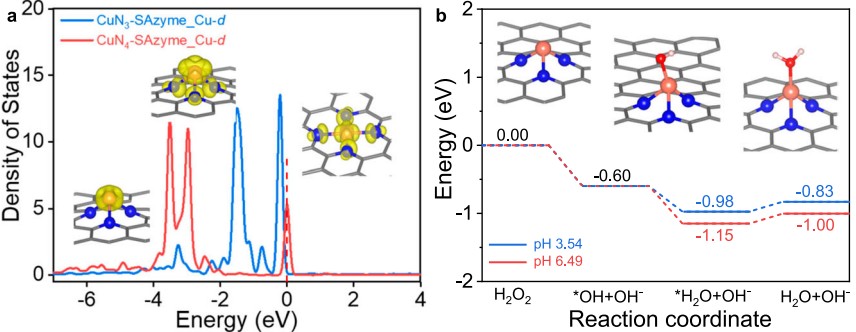

**Fig. 3 | Characterization of enzymatic activity. a** Reaction-time curves of TMB colorimetric reaction catalyzed by CuN$_x$-SAzymes and CuO$_x$ nanozymes. Inset: Photographs of peroxidase substrate (TMB, OPD, and ABTS) solutions catalyzed by CuN$_x$-SAzymes. **b** Comparison of the specific activities (U mg$^{-1}$) of CuN$_x$-SAzymes and CuO$_x$ nanozymes. **c** Evaluation of the specific activities of CuN$_3$-SAzyme under NIR light/X-ray irradiation or after irradiation by X-ray with the dose of 20 Gy/ 100 Gy (pH 3.54). **d** Enzymatic activities of CuN$_3$-SAzyme and natural HRP after irradiation by X-ray with various doses. Inset: Photographs of peroxidase substrate (TMB, OPD, and ABTS) solutions catalyzed by radiation-treated CuN$_3$-SAzyme and radiation-treated natural HRP (pH 3.54). These data are presented as mean values ± SD ($n = 3$ independent experiments).

**Fig. 4 | DFT studies on the peroxidase-like activity of CuN$_3$-SAzyme. a** Calculated density of states for the $d$-orbitals of Cu single atom on CuN$_3$-SAzyme and CuN$_4$-SAzyme. The red dashed line represents the Fermi level. **b** Calculated energy profiles for the dissociation of H$_2$O$_2$ molecule on CuN$_3$-SAzyme at two experimental pH values. Note that these produced •OH radicals are very easily involved in reactions around them, which can lead to a further decrease in the energy of the whole system. Cu, orange; C, gray; N, blue; O, red; H, pink.

DFT calculations were also carried out to investigate the enzymatic activity of CuN$_3$-SAzyme. It is found that there is a strong interaction between the Cu single atom on CuN$_3$-SAzyme and H$_2$O$_2$ molecules, making H$_2$O$_2$ dissociation being a barrierless process; while this interaction is very weak for CuN$_4$-SAzyme and thus H$_2$O$_2$ cannot be adsorbed on it. The higher activity of CuN$_3$-SAzyme than CuN$_4$-SAzyme can be interpreted in terms of both electronic and geometric factors. As shown in Fig. 4a, most of the Cu $d$-states in CuN$_3$-SAzyme lie within an energy range of 2 eV below the Fermi level, and there is a significant distribution of the $d$-states on the single Cu atom near the Fermi level. However, for the single Cu atom on CuN$_4$-SAzyme, the vast majority of its $d$-states lie outside 2 eV below the Fermi level, and the distribution

of Cu $d$-states near the Fermi level is also smaller. In addition, the single Cu atom on CuN$_3$-SAzyme protrudes more than 2 Å above the graphene plane, which can bind with H$_2$O$_2$ more easily than the in-plane Cu atom on CuN$_4$-SAzyme. Figure 4b shows the calculated energy profiles for the dissociation of H$_2$O$_2$ molecule on CuN$_3$-SAzyme at two experimental pH values (3.54 and 6.49). The cleavage of the O-O bond in H$_2$O$_2$ and the subsequent formation of the H$_2$O molecule are both readily occurring processes for CuN$_3$-SAzyme, further supporting its excellent enzymatic activity. It is worth noting that in reality the above processes are more likely to occur, because ·OH radicals produced by H$_2$O$_2$ dissociation are very easily involved in reactions in the environment where they are located, leading to a further decrease in the energy of the whole system.

## Radiation-resistance characterization of CuN$_3$-SAzyme

Environment conditions, in particular X-ray, essentially affect the enzyme activity by altering the biochemical structures[58]. Therefore, good resistance against X-ray is required for enzymes to maintain their structure and activity during radiotherapy, so that robust therapeutic efficacy would be continuously achieved to enhance therapeutic efficacy. Prior to using CuN$_3$-SAzyme as a radiosensitizer, we thus evaluated the influence of environment conditions on its peroxidase-like activity. Remarkably, high enzymatic activities (≥ 80%) are observed over a broad range of temperatures and pH levels, without substantial morphological change (Supplementary Figs. 17–19). This demonstrates that CuN$_3$-SAzyme shows better tolerance towards pH and higher stability against temperature compared to natural HRP. More interestingly, at pH 3.54, exposure of CuN$_3$-SAzyme to radiation significantly enhances its peroxidase-like activity: the specific activity and $k_{cat}/K_m$ increase to 20.24 U mg$^{-1}$ and 9.04 × 10$^3$ M$^{-1}$ min$^{-1}$ under 808 nm NIR light illumination, respectively, and to 15.81 U mg$^{-1}$ and 8.43 × 10$^3$ M$^{-1}$ min$^{-1}$ under X-ray irradiation, respectively (Fig. 3c, Supplementary Figs. 20–23, and Supplementary Table 4). These may be attributed to CuN$_3$-SAzyme-mediated photothermal effect[32,59] and X-ray-accelerated Cu$^+$/Cu$^{2+}$ conversion on CuN$_3$-SAzyme[60].

As known, X-ray is capable of damaging the structure of natural enzymes. Therefore, it is observed that the enzymatic activity of natural HRP decreases sharply after X-ray irradiation and finally approaches to zero with the radiation dose greater than 50 Gy (Fig. 3d), accompanied by the changes in its structure (Supplementary Fig. 24). For CuN$_3$-SAzyme, however, there is no significant decrease in its peroxidase-like activity or kinetic parameters after irradiation by X-ray with the radiation dose up to 100 Gy, and only a 10% decrease is detected upon further irradiation (500 Gy) (Fig. 3d, Supplementary Figs. 21 and 25, and Supplementary Table 4). Moreover, the enzymatic activity of CuN$_3$-SAzyme does not decrease even after several radiation cycles (Supplementary Fig. 26). No changes are found, either in the morphology and chemical structures of CuN$_3$-SAzyme or its enzymatic kinetics, as attested by XPS, TEM and EDS mapping (Supplementary Figs. 27–30). Furthermore, XAFS analysis shows that the coordination microenvironment of Cu atoms remains the same after X-ray irradiation (Supplementary Fig. 31). These results suggest that CuN$_3$-SAzyme is more resistant to X-ray than natural enzymes and can be efficiently reused for repeated cycles without appreciable loss of enzymatic activity. The features are crucial for various applications, in particular for radiotherapy where radiation-resistance and good recyclability are highly desirable.

This difference between CuN$_3$-SAzyme and natural HRP in terms of radioresistance against X-ray is mainly due to the following reasons. First, each active site of CuN$_3$-SAzyme contains only one Cu atom and three N atoms, and the interaction between them is sufficiently strong. This means that the CuN$_3$ active sites can maintain not only structural but also conformational stability. Additional DFT calculations show that the interaction strength between the Cu atom and the N3 group is as high as 2.96 eV, meaning that the anchored Cu atom is not easily detached from the substrate. Moreover, our ab initio molecular dynamics (AIMD) simulations for simulating the energy fluctuations of CuN$_3$-SAzyme under X-ray irradiation (using a large supercell and at a high temperature of 1,773 K) (Supplementary Fig. 32) do not find any other geometry of CuN$_3$ except the one shown in Fig. 2f, demonstrating that the CuN$_3$ active sites can be well preserved and are not susceptible to conformational changes (Supplementary Fig. 33). We note in passing that a small fraction of Cu atoms detaches from the anchoring N3 groups when the simulated temperature is further increased to 2,273 K (corresponding to more drastic energy fluctuations), which is consistent with the observed slight decrease in the peroxidase-like activity of CuN$_3$-SAzyme under high-dose irradiation (500 Gy) (Fig. 4d and Supplementary Fig. 34). When one, two, three, and four electrons are removed from CuN$_3$-SAzyme, to simulate the ionization of CuN$_3$-SAzyme under X-ray irradiation, the above results still hold (Supplementary Fig. 35), further confirming the stability of the CuN$_3$ active sites. In contrast, the active center of natural HRP contains more atoms and groups, and the interactions include not only strong chemical bonds but also weaker hydrogen bonds and intermolecular interactions. Under X-ray irradiation, these weaker interactions can be easily disrupted, causing the active center to change to other configurations and lose its original enzymatic activity. Such instabilities can be further aggravated by electron ionization caused by X-ray irradiation.

The second reason stems from the low Cu content in CuN$_3$-SAzyme, which means that the short-wavelength X-ray photons that penetrate the nanozyme have a high probability of interacting with the carbon architecture and a very low probability of interacting with the CuN$_3$ active sites directly. Previous experiments have shown that the main structure of few-layer graphene remains unchanged after X-ray irradiation[61], and our AIMD simulation results also confirm the stability of the CuN$_3$-SAzyme system (Supplementary Figs. 33 and 35). In contrast, the active center of natural HRP occupies a much larger space than that of the CuN$_3$ active moieties, and the stability of the adopted conformation of the active center also depends on its surrounding environment with an even larger size. The X-ray photons can easily hit some of the components associated with the functionality of the natural enzyme, thus destabilizing the conformation and decreasing its enzymatic activity. Taken together, these results attest that anchoring CuN$_3$ active moieties on 2D carbon nanostructures makes CuN$_3$-SAzyme much more radioresistant to X-ray and this concept could be further extended to fabricate other radioresistant SAzymes. Furthermore, it is found that CuN$_3$-SAzyme is also radioresistant to γ-ray (a cobalt-60 source, average photon energy of 1.25 MeV), without significant changes in its enzymatic activity or chemical structures (Supplementary Figs. 36-38).

## Synergistical inhibition of tumor growth by CuN$_3$-SAzyme

Encouraged by the radiation-enhanced peroxidase-like activity of CuN$_3$-SAzyme and remarkable photothermal performance (Supplementary Figs. 39, 40), we employed CuN$_3$-SAzyme as an effective therapeutics for enhanced radio-enzymatic therapy. For biomedical application, CuN$_3$-SAzyme was first cut and reduced to tiny sheets through ultrasonication, with the average hydrodynamic diameter of ~280 nm and zeta potential of -9.70 mV (Supplementary Figs. 41, 42). CuN$_3$-SAzyme can be internalized into cells via clathrin-mediated endocytosis (Supplementary Fig. 43) and triggers a concentration-dependent death, with the half maximal inhibitory concentration (IC50) of 13.12 μg mL$^{-1}$ for 4T1 cells and 18.50 μg mL$^{-1}$ for K7M2 cells (Fig. 5a and Supplementary Fig. 44). In contrast, due to lower peroxidase-like activity, CuO$_x$ nanozymes and CuN$_4$-SAzyme exhibit less toxicity at the same incubation concentration (Supplementary Fig. 45). In addition, CuN$_3$-SAzyme has higher IC50 value (23.18 μg mL$^{-1}$) for 3T3 cells as a normal model (Supplementary Fig. 46). These infer that CuN$_3$-SAzyme shows an efficient and selective

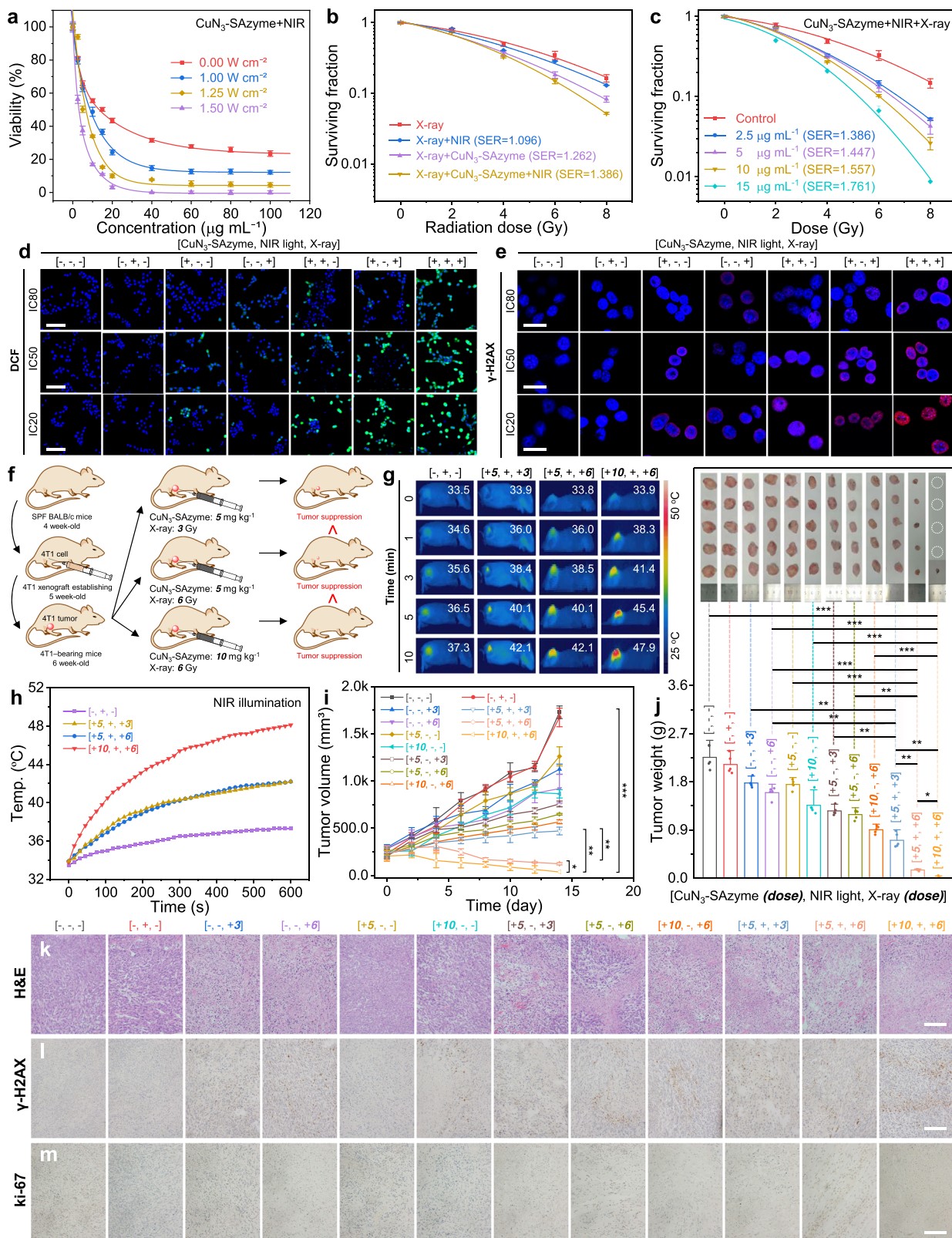

inhibiting capability for tumor cells. After exposure to 808 nm NIR light, the IC50 value of CuN3-SAzyme decreased to 8.85 μg mL⁻¹ at the power density of 1.00 W cm⁻² and further to 2.83 μg mL⁻¹ at the power density of 1.50 W cm⁻², indicating a progressive enhancement with increasing power density of 808 nm NIR light (Supplementary Fig. 47). Furthermore, CuN3-SAzyme plus co-irradiation with 808 nm NIR light and X-ray induces the best inhibition ability for colony formation and

this enhanced inhibition efficiency is concentration-dependent (Fig. 5b, c and Supplementary Figs. 48, 49). Using 2′,7′-dichloro-fluorescin diacetate as a fluorescent indicator, it is found that CuN3-SAzyme enables catalytic generation of more radical oxygen species (ROS) at the cellular level to kill cancer cells (Fig. 5d).

The ROS-related cytotoxicity of CuN3-SAzyme was further addressed by analyzing cellular response, including DNA double-

**Fig. 5 | In vitro and in vivo enhanced radio-enzymatic therapy by CuN₃-SAzyme.** **a** Viabilities of 4T1 cells after incubation with CuN₃-SAzyme and illumination by 808 nm NIR light at different power densities. **b** Colony formation curves of 4T1 cells treated with CuN₃-SAzyme (2.5 μg mL⁻¹) followed by 808 nm NIR light illumination. **c** Colony formation curves of 4T1 cells treated with CuN₃-SAzyme. **d** Fluorescence images of 4T1 cells stained with 2′,7′-dichlorofluorescin diacetate. Scale bar: 50 μm. **e** Qualitative representation of γ-H2AX foci formation of 4T1 cells. Scale bar: 25 μm. **f** Schematic illustration of the establishment of 4T1 tumor xenograft model and therapeutic outcome. **g** Representative infrared thermal images of 4T1-bearing mice. **h** Corresponding temperature rise profiles at the tumor sites. **i** Tumor growth curves of 4T1-bearing mice. **j** Weight of the dissected tumors and corresponding digital photographs. **k**–**m** Histopathology images of the excised tumors stained with H&E (**h**), γ-H2AX (**l**), and Ki-67 (**m**). Scale bar: 50 μm. Experiments were performed three times (**d**, **e**, **k**–**m**) with similar results. Representative images are presented. All data are presented as means ± SD (n = 6 independent experiments for **a**; n = 3 independent experiments for **b**, **c**; n = 5 biologically independent animals for **i**, **j**). Statistical significance was analyzed by two-tailed Student's t-test, $^*p < 0.05$, $^{**}p < 0.01$, and $^{***}p < 0.001$ (**i**, **j**).

strand breaks (DSBs) and mitochondrial membrane potential loss. When incubated with CuN₃-SAzyme, the γ-H2AX fluorescence is observed in the nuclei of 4T1 cells and sharply increases under 808 nm NIR light and/or X-ray irradiation (Fig. 5e and Supplementary Fig. 50). This indicates that the ROS generated from the elevated intracellular $H_2O_2$ indeed cause DSBs and this effect is remarkably improved upon 808 nm NIR light and X-ray irradiation. Moreover, the concentration-dependent loss of mitochondrial membrane potentials is observed in the CuN₃-SAzyme-treated 4T1 cells and the green fluorescence completely disappear when further irradiated by 808 nm NIR light and X-ray. This confirms the intensive damage of mitochondrial membrane, simultaneously accompanying with the release of cytochrome c into the cytosol and inducing mitochondria-mediated apoptosis pathway (Supplementary Figs. 51–53).

To further test the in vivo therapeutic efficacy, 4T1 mammary tumor-bearing BALB/c mice were injected intratumorally with CuN₃-SAzyme followed by X-ray irradiation (Fig. 5f). Notably, the remarkable photothermal effect enables the increase in the temperature of tumor sites (Fig. 5g, h), overcoming their tumor hypoxia microenvironment (Supplementary Figs. 54, 55). According to the variation in the tumor volume, co-treatment with CuN₃-SAzyme and X-ray leads to an enhanced tumor inhibition efficacy on tumor growth at 5 and 10 mg kg⁻¹ doses (Fig. 5i, j). The highest suppression effect is obtained in the CuN₃-SAzyme treated group plus co-irradiation and also confirmed by the evidence from the hematoxylin and eosin (H&E)- and γ-H2AX-stained histopathological images, as well as the lowest antigen Ki-67 expression (Fig. 5k–m and Supplementary Figs. 56, 57). These results demonstrate that CuN₃-SAzyme could effectively suppress 4T1 tumor growth in vivo mainly due to irradiation-enhanced peroxidase-like activity which enables the generation of more ROS from endogenous $H_2O_2$ and to the revision of the hypoxic microenvironment via CuN₃-SAzyme-mediated photothermal effect.

Finally, the safety profile of CuN₃-SAzyme was evaluated by examining the loss of body weight after intratumoral and subcutaneous injection in mice. As expected, CuN₃-SAzyme exhibits good hemocompatibility without red blood cell damage and induces no significant variations in body weight at various doses (5 and 10 mg kg⁻¹ for intratumoral injection, 10 and 20 mg kg⁻¹ for subcutaneous injection) (Supplementary Figs. 58–60). Except in tumors, no systemic inflammation or tissue damage are observed in the major organs such as the heart, liver, spleen, lung, and kidney at day 14 post-injection (Supplementary Figs. 61–62). Moreover, the index levels from blood routine test and blood biochemistry test fall within the reference ranges of the untreated mice (Supplementary Figs. 63–64). In addition, the tissue distribution of CuN₃-SAzyme in mice following subcutaneous injection does not change over time between organs and CuN₃-SAzyme are mainly located at the injection site (Supplementary Fig. 65). These preliminary data demonstrate that CuN₃-SAzyme is safe for local administration and can be used as an effective therapeutic agent for cancer treatment.

## Discussion

In summary, we theoretically demonstrated and experimentally fabricated a CuN₃-SAzyme with higher enzymatic activity compared to CuN₄-SAzyme through modulating Cu-Nₓ coordination structures. The introduction of external fields, including 808 nm NIR light and X-ray, can remarkably enhance the enzymatic activity and kinetics of CuN3-SAzyme. Moreover, CuN₃-SAzyme is rather stable after repeated X-ray irradiation and high-dose irradiation, without a significant decrease in its enzymatic activity, and thus displays superior radioresistance against X-ray. As a result, CuN₃-SAzyme can completely eradicate cancerous cells through enhanced radio-enzymatic modality and significantly decrease the damage to normal tissues. Such a simple and valuable coordination engineering strategy not only provides insights into the development and optimization of SAzymes with excellent enzymatic activity and high radioresistance, but also can be extended to therapeutics for various diseases, such as tumors by enhanced radio-enzymatic therapy, thereby opening up new promising and breakthroughs in enzymatic applications.

## Methods
### Ethical statement
All animal experiments were performed in accordance with the published guidelines of the CAS Key Laboratory for Biomedical Effects of Nanomaterials and Nanosafety (Institute of High Energy Physics and National Center for Nanoscience and Technology; Approval ID: IHEPLLSC-2023-52). All mice were housed in a specific pathogen-free environment on a 12/12 h light-dark cycle with the standard conditions: Temperature, 20-25 °C; Relative humidity, 40-70%. All research was carried out according to relevant guidelines and regulations. The maximal tumor size permitted by the Animal Experiment Administration Committee of the Institute of High Energy Physics and National Center for Nanoscience and Technology is 15 mm in diameter, and no mice exceeded this criterion in this work.

### Materials
Copper (II) chloride hydrate (CuCl₂·H₂O, 99 + %,), ascorbic acid (AA, 99 + %), copper nitrate hydrate (Cu(NO₃)₂·xH₂O, 99 + %), zinc nitrate hexahydrate (Zn(NO₃)₂·6H₂O, 99 + %), 2-methylimidazole (98%), and glacial acetic acid (99%) were purchased from Alfa Aesar. Copper(I) acetate (CuCH₃COO, 97%) and Tween-80 were purchased from Sigma-Aldrich. Other chemical reagents, including sodium hydroxide solution, potassium chloride, methanol, and ethanol, were supplied from Beijing Chemical Reagent Co. All reagents were used without further purification.

### Synthesis of CuN₃-SAzyme
0.7275 g of Cu(NO₃)₂·xH₂O and 0.8210 g of 2-methylimidazole were dissolved in absolute methanol (30 mL), respectively. Upon mixing with each other, 500 g of KCl was added in the resulting solution. The precipitate was collected by centrifugation (7162 g, 5 min) and annealed at 850 °C for 2 h under argon flow. Finally, the sample was washed with $H_2O$ and $H_2SO_4$ solution several times. Similarly, Cu-free N-doped carbon support was prepared in the absence of Cu(NO₃)₂·xH₂O.

### Synthesis of CuN₄-SAzyme
First, ZIF-8 was synthesized. 2.6260 g of 2-MI and 2.2380 g of Zn(NO₃)₂·6H₂O were dissolved in absolute methanol (30 mL), respectively, and then mixed with each other at room temperature. The

resulting precipitate was centrifuged (7,162 g, 5 min) and washed with absolute methanol several times. After dried in vacuum at 70 °C overnight, the ZIF-8 powder was placed in a tube furnace and heated at 900 °C for 3 h under flowing argon gas. The product was cooled down to room temperature naturally (named carbon nanoparticles, CNs).

Second, 0.0100 g of $Cu(NO_3)_2 \cdot xH_2O$ and 2.0000 g of urea were dissolved in absolute methanol (30 mL) and then mixed the resulting NCs (200 mg). When the mixture was ultra-sonicated for 1 h and then stirred at room temperature for 6 h, the resulting precipitate was centrifuged (7162 g, 5 min) and washed with absolute methanol several times. The dried powder was transferred into a ceramic boat and placed in a tube furnace to be heated at 700 °C for 3 h under flowing argon gas, the sample was naturally cooled down to room temperature and washed with $H_2O$ and $H_2SO_4$ several times.

## Synthesis of $Cu_2O$ nanozyme

Typically, 1 mmol $CuCl_2 \cdot H_2O$ was dissolved in the deionized water (100 mL) under vigorous stirring. The obtained solution was placed into an oil bath and heated at 50 °C for 0.5 h, followed by the addition of 2 M NaOH solution (10 mL). After stirring at 50 °C for another 0.5 h, 0.6 M AA solution (10 mL) was added dropwise (3 mL min⁻¹) in the dark brown solution. This red mixture was aged at 50 °C for 3 h and finally cooled down to room temperature naturally. The resulting red precipitate was collected by centrifugation (13,400 g, 5 min) and washed with deionized water several times to remove the residuals. The precipitate was re-dispersed in deionized water and the concentration of this dispersion was measured by inductively coupled plasma mass spectrometry (ICP-MS, Thermo-X7, USA).

## Synthesis of CuO nanozyme

In brief, 1 mmol $CuCH_3COO$ was dispersed in the deionized water (50 mL) under sonication and 0.25 mL of glacial acetic acid was added dropwise (0.25 mL min⁻¹) in the resulting solution. After heating to 100 °C under vigorous stirring, 1 M NaOH solution (5 mL) was added to form the black precipitate. This black precipitate was separated by centrifugation (13,400 g, 5 min) and washed with deionized water several times. The concentration of this dispersion was measured by ICP-MS.

## Characterization

Transmission electron microscopy images and elemental mapping were obtained from a Tecnai $G^2$ F20 microscope operated at 200 kV (FEI, USA) and equipped with an energy-dispersive X-ray analysis system. High-angle annular dark-field scanning transmission electron microscopy (HAADF-STEM) was performed on a JEOL-ARM300F microscope equipped with a spherical aberration corrector at 200 keV. X-ray photoelectron spectroscopy (XPS) analysis was performed on a VG Multilab 2000 instrument (Thermo Fisher). X-ray diffraction (XRD) patterns were obtained from a D8 ADVANCE apparatus (Bruker, Germany; Cu $K_\alpha$, 0.15406 nm). Inductively coupled plasma optical emission spectroscopy (ICP-OES) was conducted using an Agilent 720ES apparatus. X-ray absorption fine structure (XAFS) spectra at Cu K-edge were acquired at the 1W1B station in Beijing Synchrotron Radiation Facility (BSRF) and the BL11B station in Shanghai Synchrotron Radiation Facility (SSRF).

## XAFS measurement and analysis

The Cu K-edge XAFS data were recorded in the fluorescence mode, using Cu foil and $CuO_x$ nanozymes as references. All spectra were collected under ambient conditions. The extended Cu K-edge EXAFS data were processed using the ATHENA module in the IFEFFIT software packages. To obtain $k^3$-weighted EXAFS spectra, the post-edge background was subtracted from the overall absorption and then normalized to the edge-jump step. These $k^3$-weighted $\chi(k)$ data of Cu K-edge were Fourier transformed into real (R) space using a Hanning window

(dk = 1.0 Å⁻¹) to isolate the EXAFS contributions from various coordination shells. Quantitative structural parameters around the central atoms were derived through least-squares curve fitting, performed with the ARTEMIS module of the IFEFFIT software packages, using the following EXAFS equation:

$$\chi(k) = \sum_j \frac{N_j S_0^2 F_j(k)}{k R_j^2} \exp\left[-2k^2\sigma_j^2\right] \exp\left[\frac{-2R_j}{\lambda(k)}\right] \sin[2kR_j + \phi_j(k)] \quad (1)$$

where, $S_0^2$ is the amplitude reduction factor, $F_j(k)$ is the effective curved-wave backscattering amplitude, $N_j$ is the number of neighbors in the $j^{th}$ atomic shell, $R_j$ is the distance between the X-ray absorbing central atom and the atoms in the $j^{th}$ atomic shell (backscatterer), $\lambda$ is the mean free path in Å, $\phi_j(k)$ is the phase shift (including the phase shift for each shell and the total central atom phase shift), $\sigma_j$ is the Debye-Waller parameter of the $j^{th}$ atomic shell (variation of distances around the average $R_j$). The functions $F_j(k)$, $\lambda$, and $\phi_j(k)$ were calculated with the ab initio code FEFF8.2. The additional details for the EXAFS simulations are given below. The coordination numbers of the model samples were set to their nominal values, and the obtained $S_0^2$ was kept constant in subsequent fittings. However, the internal atomic distances (R), Debye-Waller factor ($\sigma^2$), and the edge-energy shift ($\Delta E_O$) were allowed to run freely.

## Measurement of photothermal effect of $CuN_3$-SAzyme

The aqueous dispersions (1 mL) of $CuN_3$-SAzyme with different concentrations were exposed to 808 nm NIR light (1.00 W cm⁻²) for 10 min. At each interval time, the temperature and infrared thermal images were recorded by the thermal camera (FLIR Therma CAM E40). To evaluate the influence of power density on the temperature change, 50 μg mL⁻¹ of the aqueous dispersions of $CuN_3$-SAzyme were illuminated with 808 nm NIR light with the power density ranging from 0.25 W cm⁻² to 2.00 W cm⁻² and the temperature was recorded by the same protocol as described above. To calculate the photothermal conversion efficiency of $CuN_3$-SAzyme, 25 μg mL⁻¹ of the aqueous dispersion of $CuN_3$-SAzyme was illuminated with 808 nm NIR light at the power density of 1.00 W cm⁻² for 750 s and then cooled naturally for 1500 s, respectively. 1 mL of pure deionized water was set as a control sample. To assess the photothermal stability under 808 nm NIR light (1.00 W cm⁻²), the aqueous dispersion of $CuN_3$-SAzyme was illuminated with five heating/cooling (600 s/600 s) cycles. The stability of $CuN_3$-SAzyme in the physiological solutions was also detected by monitoring the heating profiles of $CuN_3$-SAzyme after storage for various days.

## Measurement of enzyme-like activities of $CuN_x$-SAzymes and $CuO_x$ nanozymes

The peroxidase-like property was detected by colorimetric assays. In a typically procedure, 3.6 μL of aqueous suspension of $CuN_x$-SAzyme (1.0 mg mL⁻¹), 30 μL of TMB solution (6 mM), and 22.5 μL of $H_2O_2$ solution (400 mM) were mixed into HAc-NaAc buffer solution (10 mM, pH 3.54) with a final volume of 180 μL. The catalytic oxidation of TMB (ox-TMB) was studied by monitoring the absorbance of ox-TMB (652 nm, $\varepsilon$ = 39,000 M⁻¹ cm⁻¹) using a microplate spectrophotometer (Multiskan MK3, Thermo Fisher Scientific, USA). As for the kinetic data towards TMB substrate, TMB solutions with different concentrations were added into HAc-NaAc buffer solutions containing $CuN_x$-SAzyme (3.6 μL, 1 mg mL⁻¹) and $H_2O_2$ (22.5 μL, 400 mM), finally obtaining the mixtures with the final volume of 180 μL. The kinetic data towards $H_2O_2$ substrate was obtained using the method as described above except the addition of $H_2O_2$ solution with different concentrations into HAc-NaAc buffer solutions containing $CuN_x$-SAzyme (3.6 μL, 1.0 mg mL⁻¹) and TMB (30 μL, 6 mM). The kinetic data were calculated

using the following typical Michaelis-Menten equation,

$$v = \frac{V_{max} \cdot [S]}{K_M + [S]} \tag{2}$$

where $v$ is the initial velocity, $[S]$ is the concentration of the substrate, $K_M$ is the Michaelis-Menten constant, and $V_{max}$ is the maximal reaction velocity. The specific activities (U mg$^{-1}$) were measured by monitoring the absorbance of the mixture (the total volume was 180 μL) containing 60 μL of H$_2$O$_2$ solution (9 M), 30 μL of TMB solution (6 mM), and various concentrations of CuN$_x$-SAzyme. In particular for catalase-like enzymatic activity, the concentration of generated O$_2$ was detected by a specific oxygen electrode (JPSJ-605F, INESA). Meanwhile, the external field-enhanced peroxidase-like performance was detected using the same protocol besides the introduction of X-/γ-ray and 808 nm NIR light (1.00 W cm$^{-2}$).

## Evaluation of the stability of CuN$_3$-SAzyme

To assess the influence of temperature and pH on the enzymatic-like performance, the absorbance of the mixture (the total volume was 180 μL) containing 3.6 μL of aqueous suspension of CuN$_x$-SAzyme (1.0 mg mL$^{-1}$), 30 μL of TMB solution (6 mM), and 22.5 μL of H$_2$O$_2$ solution (400 mM) was recorded in the range of temperature from 7 °C to 87 °C and pH value from 2.16 to 12.04. For the assessment of radio-resistance, the aqueous suspensions of CuN$_3$-SAzyme were irradiated by X-/γ-ray with the radiation dose ranging from 1 Gy to 500 Gy, and then the enzymatic-like activity was measured using the protocol as described above.

## Evaluation of cytotoxicity

Luciferase-transfected murine breast carcinoma cell line (4T1-Luc), mouse osteosarcoma cell line (K7M2), and mouse embryonic fibroblast cell line (3T3) were obtained from Peking Union Medical College Hospital. In particular, 4T1-Luc cell line was authenticated using Short Tandem Repeat (STR) analysis on Apr 15th 2024. The scientific justification demonstrated that 4T1-Luc cells are not cross-contaminated or otherwise misidentified cell lines (Supplementary Table 5). Cells were cultured in Dulbecco's Modified Eagle Medium (DMEM, Gibco, USA) containing 10% Fetal Bovine Serum (FBS, Gibco, USA) and 1% penicillin-streptomycin (Gibco, USA) in 5% CO$_2$ at 37 °C. When seeded in the 96-well plates for 24 h (6,000 cells/well for 4T1-Luc cells; 6,000 cells/well for K7M2 cells; 6,000 cells/well for 3T3 cells), cultured cells were co-incubated with CuN$_3$-SAzyme, CuN$_4$-SAzyme, CuO nanozyme or Cu$_2$O nanozyme at various concentrations for 24 h (consistent with the content of copper in CuN$_3$-SAzyme). The standard Cell Counting Kit-8 (CCK-8, Dojindo, Japan) assay was performed to determine the cytotoxicity via recording the absorbance at 450 nm using the microplate spectrophotometer. To evaluate photothermal cytotoxicity, CuN$_3$-SAzyme-treated cells were further illuminated with 808 nm NIR light at various power densities for 10 min and the cell viability was measured using the same protocol as described above.

## Live/dead cell staining assay

To further evaluate the photothermal killing efficacy, 4T1-Luc cells were seeded in 6-weel plates for 24 h (200,000 cells/well) and treated with of CuN$_3$-SAzyme at different concentrations. After co-incubation for 6 h, these treated cells were illuminated with 808 nm NIR light at different power densities for 10 min. Following co-incubation for another 18 h, these cells were stained with calcein AM (CA, Beyotime, China) and propidium iodide (PI, Beyotime, China) for 20 min at 37 °C in the dark. Finally, the fluorescent images were collected using an inverted fluorescence microscope (Olympus X73, Tokyo, Japan).

## Colony formation assay

To evaluate the inhibition effects, 4T1-Luc cells seeded in 6-well plates (2,000 cells/well) were incubated with CuN$_3$-SAzyme. After incubation for 6 h, these treated cells were irradiated with 808 nm NIR light (1.00 W cm$^{-2}$) for 10 min and/or X-ray (6 Gy). Following incubation for another 18 h, these cells were continued to be cultured in fresh DMEM medium for 7 days. Subsequently, formed clones were fixed with 4% paraformaldehyde (Innochem, China) for 10 min and stained by Giemsa Staining Solution (Beyotime, China) for 30 min. Finally, clones were counted and used to plot the survival fraction. To further evaluate the radiosensitization efficiency of CuN$_3$-SAzyme, 4T1-Luc cells were incubated with CuN$_3$-SAzyme at different concentrations for 6 h and then irradiated by X-ray with different doses for 10 min and/or 808 nm NIR light (1.00 W cm$^{-2}$) for another 10 min. Following incubation for 7 days, the number of clones were obtained using the same protocol as described above to plot the survival fraction.

## Detection of intracellular ROS

To detect the level of intracellular ROS, 4T1-Luc cells were seeded in the confocal dish (200,000 cells/well) and treated with CuN$_3$-SAzyme at various concentrations for 6 h. When irradiated with 808 nm NIR light (1.00 W cm$^{-2}$) for 10 min and/or X-ray (6 Gy) for another 10 min, these treated cells were co-stained with 2′,7′-dichlorofluorescin diacetate (DCFH-DA, Beyotime, China) and the nuclei probe 2-(4-Amidinophenyl)-6-indolecarbamidine dihydrochloride (DAPI, Beyotime, China) for 20 min at 37 °C in the dark. The fluorescent images were recorded using a fluorescence confocal microscope (A1/LSM-Kit, Nikon/PicoQuant GmbH, Japan/Germany).

## Evaluation of intracellular mitochondrial membrane potential

To detect the change of intracellular mitochondrial membrane potential (MMP), 4T1-Luc cells (200,000 cells/well) were treated with CuN$_3$-SAzyme at different concentrations for 6 h and then irradiated with 808 nm NIR light (1.00 W cm$^{-2}$) for 10 min and/or X-ray (6 Gy) for another 10 min. After incubation for 12 h, these treated cells were incubated with JC-1 working solution (10 μg mL$^{-1}$) for 20 min. Finally, the change of intracellular MMP was monitored by recording the fluorescent images using the fluorescence confocal microscope.

## Evaluation of intracellular DNA damage

To detect the intracellular DNA damage, CuN$_3$-SAzyme-treated cells (20,000 cells/well) were irradiated with 808 nm NIR light (1.00 W cm$^{-2}$) for 10 min and/or X-ray (6 Gy) for another 10 min. After incubation for 12 h, these cells were fixed with 4% paraformaldehyde for 10 min and treated with 0.25% Triton X-100 to enhance the permeability of cell membranes. Following the blocking with 1% bovine serum albumin, these treated cells were labeled by γ-H2AX mouse monoclonal antibody overnight and secondary antibody of Cy3-conjugated sheep anti-mouse for 2 h. The intracellular DNA damage was monitored by recording the fluorescent images using the fluorescence confocal microscope.

## Evaluation of cytochrome c release

In detail, 4T1-Luc cells were pre-seeded in 24-well plates (20,000 cells/well) and then treated with CuN$_3$-SAzyme. After incubation for 6 h, these cells were irradiated with 808 nm NIR light (1.00 W cm$^{-2}$) for 10 min and/or X-ray (6 Gy). After another 12 h, these treated cells were stained by MitoTracker Red CMXRos for 30 min and fixed with 4% paraformaldehyde for another 10 min. Subsequently, 0.5% Triton X-100 was used to permeabilize the cell membrane. When further blocked by 1% bovine serum albumin for 1 h at room temperature and labeled with Cyt c antibody (1:1000) overnight at 4 °C, these cells were co-labeled with Alexa Fluor 488 secondary antibody and nuclei probe DAPI. Finally, the fluorescent images were recorded using the fluorescence confocal microscope.

## Western blot analysis apoptosis-related proteins

After different treatments, 4T1-Luc cells (1,000,000 cells/well) were collected and lysed with radioimmunoprecipitation assay buffer (50 mM Tris-HCl containing 1% NP-40, 1 mM EDTA, 0.1% SDS, 150 mM NaCl, supplemented with 1 mM PMSF, pH 7.50) on ice for 30 min. The whole protein of cells was obtained by centrifugation at 4 °C for 15 min. The concentration of whole protein was quantified by the BCA Protein Assay Kit (Beyotime, China). Subsequently, the apoptosis-related proteins were separated by sodium dodecyl sulfate-polyacrylamide gel electrophoresis and blotted on nitrocellulose membranes. The primary antibody was incubated with target protein at 4 °C overnight. After using Tris-buffer saline (50 mM Tris-HCl, pH 7.5, 150 mM NaCl, 1‰ Tween-20) to wash the nitrocellulose membranes three times, the secondary antibody was incubated for 2 h. The protein marker and target protein undergo electrophoresis on the same gel under identical conditions. The molecular weight of target protein of interest by simply comparing the position of its band on the gel with those of the PageRuler prestained protein marker (Catalog number. 26616, Thermo Fisher Scientific, US). The target protein visualization was performed by enhanced chemiluminescence (ECL, Beyotime, P0018).

## In vitro apoptosis analysis

4T1-Luc cells were pre-seeded in 6-well plates (200,000 cells/well) and incubated with CuN$_3$-SAzyme for 6 h and sequentially irradiated with 808 nm NIR light (1.00 W cm$^{-2}$) for 10 min and/or X-ray (6 Gy). After another 24 h, these treated cells were collected and stained by annexin V-FITC (AV, Dojindo, Japan) and propidium iodide (PI, Dojindo, Japan) at 37 °C. Finally, the apoptosis cells were quantified by flow cytometry (BD Accuri C6, USA).

## Cellular uptake pathway of CuN$_3$-SAzyme

Typically, 4T1-Luc cells were seeded in 6-well plates (200,000 cells/well) and incubated with various inhibitors including filipin III (7.5 μM), chlorpromazine (50 μM), cytochalasin D (5 μM), and wortmannin (5 μM) for 30 min. Then, FITC-labeled CuN$_3$-SAzyme was added and incubated for another 2 h. Finally, cells were washed three times with phosphate buffer saline (PBS, pH 7.40) and analyzed immediately using flow cytometry.

## Hemolysis assay

Fresh mouse blood was obtained from BALB/c mice (6-8 weeks old, female) and washed with PBS three times to collect mouse blood cells (RBCs). The resulting RBCs were re-dispersed in aqueous suspensions of CuN$_3$-SAzyme with different concentrations. The deionized water was used as the corresponding positive control and PBS (pH 7.40) was used as the negative control. After incubation for 4 h, the specimens were centrifuged at 2990 g for 5 min and the absorbance of the supernatant was detected using the microplate spectrophotometer. The hemolysis ratios of CuN$_3$-SAzyme were calculated by the following equation:

$$\text{Hemolysis ratio} = \frac{\text{OD(test)} - \text{OD(negative control)}}{\text{OD(positive control)} - \text{OD(negative control)}} \quad (3)$$

## Evaluation of enhanced radio-enzymatic therapy.

BALB/c mice (4-6 weeks old, female) were provided by Beijing HFK Bioscience Co., Ltd. To establish the 4T1 bear tumor model, 4T1 cells suspended in PBS were subcutaneously injected into the right flank of the mice. Tumor volumes were calculated by the following equation:

$$V = \frac{L \times W^2}{2} \quad (4)$$

where L (mm) is the tumor long dimension and W (mm) is the tumor width.

When the tumor volume reached a size of about 100 mm$^3$, the mice were randomly divided into several groups ($n = 5$). After intratumoral injection of CuN$_3$-SAzyme dispersed in 5% glucose solution (volume: 50 μL; dose: 5 mg kg$^{-1}$ and 10 mg kg$^{-1}$), mice were irradiated with 808 nm NIR light (1.00 W cm$^{-2}$) for 10 min and/or X-ray (3 Gy or 6 Gy). Tumor volume and body weight of mice were measured every other day. At the end of treatment period, the tumors from different groups were exfoliated, weighed, and fixed with 4% polyoxymethylene (v/v). The exfoliated tumors were sliced to hematoxylin and eosin (H&E, Abcam, UK) staining, immunohistochemical analysis of Ki-67 (Beyotime, China) and γ-H2AX (Beyotime, China).

## Evaluation of biosafety of CuN$_3$-SAzyme

BALB/c mice (6-8 weeks old, female) were subcutaneously injected by CuN$_3$-SAzyme with various doses. At each interval time, treated mice were sacrificed and major organs including the heart, liver, spleen, lung, and kidney were dissected and harvested. To prepare the ICP-MS samples, the harvested organs were weighed and mixed with nitric acid (65%, m/m) and incubated at 60 °C for 72 h. After dilution and filtration, the biodistribution of Cu element was measured by ICP-MS and calculated as Cu percentage over administrated dose per gram of tissues.

## In vivo photoacoustic imaging

4T1-bearing BALB/c nude mice (5–6 weeks old, male) were anesthetized by inhalation of isoflurane/air for 5 min. Subsequently, 25 μL of the s4T1-bearing BALB/c nude mice (5–6 weeks old, male)uspension of CuN$_3$-SAzyme (4 mg mL$^{-1}$) was intratumorally injected. The mice were illuminated by 808 nm NIR light (1.00 W cm$^{-2}$) for 10 min. The photoacoustic images were collected and analyzed on a multispectral optoacoustic tomography (MSOT, iThera Medical 128, Germany) system (750 nm to 900 nm).

## Computational details

The structural relaxations, energy calculations, and electronic structure analyzes were performed using the Vienna ab initio simulation package (VASP)[62,63]. The interactions between ionic cores and valence electrons were considered by using the projector augmented wave (PAW) method[64], and the exchange-correlation interactions were described by the optPBE-vdW functional[65,66]. The energy cutoff for the plane-wave basis sets was set to 500 eV. Both CuN$_3$ and CuN$_4$ units were embedded in a 5 × 5 graphene unit cell. The vacuum space between consecutive slabs along the vertical direction was set to 14 Å. A 3 × 3 × 1 Monkhorst-Pack grid was utilized for the first Brillouin zone sampling[67]. The atomic coordinates were relaxed until the maximum force was less than 0.03 eV Å$^{-1}$. All structures were visualized using the program VESTA[68].

Under irradiation, X-ray excites electrons in the inner layers of constituent atoms, causing not only radiative but also non-radiative transitions. The non-radiative transitions could induce a drastic energetic stimulation and lead to further ionization of the matter. Considering that it is rather complex to directly simulate the structural evolution of matter under X-ray radiation from first principles, we focused on whether CuN$_3$-SAzyme can maintain its stability under drastic energy fluctuations (equivalent to being in a hot environment of thousands of Kelvin) and when one to several electrons are ionized, by using extensive ab initio molecular dynamics (AIMD) simulations. The AIMD simulations were performed using the CP2K package[69]. The unit cell used here is nine times (3 × 3) the size of the unit cell used in the VASP calculations, containing a total of 414 C atoms, 27 N atoms, and 9 Cu atoms. The exchange-correlation interactions were described by the PBE functional[70], and the empirical scheme (D3) developed by Grimme et al.[71] was employed for the dispersion correction. We used the optimized double-ξ Gaussian basis sets[72] and an auxiliary plane wave basis set (energy cutoff: 500 Ry)[73] to expand wavefunctions. The

core electrons were treated with the scalar relativistic norm-conserving pseudopotentials[74] and the number of valence electrons adopted for C, N, and Cu were 4, 5, and 11, respectively. Only the Γ point was used for the first Brillouin zone sampling. The canonical (NVT) ensemble was adopted in the AIMD calculations with a time step of 1 fs. The Nosé-Hoover thermostats[75,76] were employed here and the corresponding temperature was set to a range between 1773 K and 3273 K (equivalent to 1,500 °C and 3000 °C, respectively), to simulate potential drastic energy fluctuations that the CuN$_3$-SAzyme may experience under X-ray irradiation.

### Reporting summary

Further information on research design is available in the Nature Portfolio Reporting Summary linked to this article.

## Data availability

The data generated in this study are present in the main text and the Supplementary Information file. Source data are provided with this paper.

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

## Acknowledgements

The authors gratefully acknowledge the Basic Science Center Project of the National Natural Science Foundation of China (22388101, Y.Z.; 22388102, Y.L.), the National Natural Science Foundation of China (22222608, L.Y.; 21890383, D.W.; 21871159, D.W.; 22171157, D.W.; 22273091, Q.F.), the National Key R&D Program of China (2018YFA0702003, Y.L.), the Chinese Academy of Sciences Project for Young Scientists in Basic Research (YSBR-054, Q.F.), the Innovation Program for Quantum Science and Technology (2021ZD0303302, Q.F.), the Beijing Natural Science Foundation (No. 2224103, J.W.) and China Postdoctoral Science Foundation (2021M691759, J.W.; 2021M700981, Q.L.). We thank the BL11B station at the Shanghai Synchrotron Radiation Facility and the 1W1B and 4B7A stations at the Beijing Synchrotron Radiation Facility. Q.F. thanks the USTC supercomputing center for providing computational resources for this project.

## Author contributions

Y.L., D.W., and Y.L. conceived the idea and designed the research project. J.W. and X.Z. designed the synthesis and performance experiments, collected and analyzed the data, and wrote the manuscript. Q.L. contributed to the characterizations of samples and wrote the manuscript.

Q.F. and B.W. contributed to the computational results and wrote the manuscript. B.L., S.W., Q.C., H.X., and C.Y. designed the characterizations experiments and analyzed the data. Q.Q.L., L.H., Y.L., D.W., Y.Z., and Y.L. contributed to revising the manuscript. All the authors commented on the manuscript and have given approval to the final version of the manuscript. J.W. and X.Z. contributed equally.

## Competing interests

The authors declare no competing interests.

## Additional information

[1]Department of Chemistry, Tsinghua University, Beijing 100084, China. [2]CAS Key Laboratory for Biomedical Effects of Nanomaterials and Nanosafety, Institute of High Energy Physics and National Center for Nanoscience and Technology, Chinese Academy of Sciences, Beijing 100049, China. [3]Institute of Marine Science and Technology, Shandong University, Qingdao 266237, China. [4]Wuhan National Laboratory for Optoelectronics, Huazhong University of Science and Technology, Wuhan 430074, China. [5]School of Future Technology, University of Science and Technology of China, Hefei 230026, China. [6]Hefei National Laboratory, University of Science and Technology of China, Hefei 230088, China. [7]Institute of Quality Standards & Testing Technology for Agro-Products, Chinese Academy of Agricultural Sciences, Beijing 100081, China. [8]GBA Research Innovation Institute for Nanotechnology, Guangdong 510700, China. [9]College of Chemistry, Beijing Normal University, Beijing 100875, China. [10]The Key Laboratory of Functional Molecular Solids, Ministry of Education, College of Chemistry and Materials Science, Anhui Normal University, Wuhu 241002, China. [11]These authors contributed equally: Jiabin Wu, Xianyu Zhu. ✉e-mail: qfu3@ustc.edu.cn; yanliang@ihep.ac.cn; ydli@mail.tsinghua.edu.cn

