## [Peer Review File · Nature Communications]

Reviewers' Comments:

Reviewer #1:

Remarks to the Author:

The authors designed and synthesized an efficient single-atom nanozyme that consists of densely isolated CuN3 sites, but not CuN4 sites by using KCl crystal as the template, for catalytic decomposition of H₂O₂ to yield hydroxyl radicals. Via regulating local coordination number, the resultant CuN3-SAzyme exhibited higher peroxidase-like activity than CuN4-SAzyme as well as CuO_x nanozymes. This was fully demonstrated by density functional theory calculation. Moreover, this carbon-based nanosheets could stabilize the geometrical structure of CuN3 sites and maintain the enzyme activity against X-rays. The changes in structure and morphology of CuN3-SAzyme after repeated irradiation by X-rays were thoroughly investigated using HAADF-STEM and XAFS measurements. Based on these advantages, this CuN3-SAzyme was further applied to overcome the shortcomings of natural enzyme in the application of radiotherapy and would enhance the therapeutic efficacy of radio-enzymatic therapy both in vitro and in vivo.

This manuscript presents new presented intriguing results based on single-atom Cu nanozyme. Of note, this work not only provides an important protocol to design single-atom nanozymes with high catalytic activity and excellent radio-resistance for various applications, for example, radiotherapy, but also offers an alternative guide to other fields including materials science, nanotechnology, and so on. Overall, I strongly recommend its publication on Nature Communications after addressing the following comments.

1. Keep the abbreviations in consistent, for example, "Fourier transformed EXAFS" in page 19 and "FT-EXAFS" in page 6.
2. Additional measurements should be provided to further demonstrate the structural stability of CuN3-SAzyme after X-ray irradiation, such as atomic-level HAADF-STEM.
3. How about XPS spectra of CuN3-SAzyme after X-ray irradiation and testing its enzymatic activity?
4. The authors could conduct an in-depth study on the cellular uptake of CuN3-SAzyme.
5. Does CuN3-SAzyme have comparable killing effects on other types of tumor cells, such as osteosarcoma K7M2 cells?
6. As demonstrated, CuN3-SAzyme showed significant cytotoxicity to 4T1 cells. However, what about the cytotoxicity of CuN4-SAzyme and CuO_x nanozymes?
7. As previously reported (Advanced Materials. 2022, 2205324), nanozymes exhibited catalase-like activity by catalyzing H₂O₂ to generate O₂, which could relieve hypoxia and enhance the effect of radiotherapy. How about the catalase-like activity of CuN3-SAzyme, does this activity affect the therapeutic efficacy?
8. Besides long-term toxicity, short-term toxicity of CuN3-SAzyme would be also assessed.

Reviewer #2:

Remarks to the Author:

The manuscript entitled "Enhancing Radiation-resistance and Peroxidase-like Activity of Single-atom Copper Nanozyme via Local Coordination Manipulation" theoretically and experimentally demonstrated that a CuN3-SAzyme shows excellent catalytic performance and also irradiation resistance. Furthermore, CuN3-SAzyme performs great effect on eradicating cancerous cells without seriously damage to normal tissues. The manuscript is technically sound and the logic is fine. However, I have some questions and suggestions to authors before I can give a judgement on this manuscript.

1. "Synthesis and Characterization of CuN_x-SAzymes" section. I don't think XPS can provide the element distribution. I don't think this characterization is necessary.
2. In Fig.1 e-g. How the authors confirm the bright dots are Cu atoms but not other contaminations? Is there any possible to give the strong evidence to prove they are single Cu atoms through TEM?

3. "Radiation-resistance characterization of CuN3-SAzyme." Section. The authors should provide clear background and knowledge to readers that why this catalytic needs to be radiation resistance under X-ray and NIR. What about other radiation source, such as α ? γ ?
4. The most drawback of this manuscript is the mechanism of radiation-resistance of this catalytic is not clear. Why this structure of catalytic has strong effect to radiation-resistance? The mechanisms of the resistance need to be studied in depth.

Reviewers' Comments:

Reviewer #1:

Remarks to the Author:

The authors designed and synthesized an efficient single-atom nanozyme that consists of densely isolated CuN₃ sites, but not CuN₄ sites by using KCl crystal as the template, for catalytic decomposition of H₂O₂ to yield hydroxyl radicals. Via regulating local coordination number, the resultant CuN₃-SAzyme exhibited higher peroxidase-like activity than CuN₄-SAzyme as well as CuO_x nanozymes. This was fully demonstrated by density functional theory calculation. Moreover, this carbon-based nanosheets could stabilize the geometrical structure of CuN₃ sites and maintain the enzyme activity against X-rays. The changes in structure and morphology of CuN₃-SAzyme after repeated irradiation by X-rays were thoroughly investigated using HAADF-STEM and XAFS measurements. Based on these advantages, this CuN₃-SAzyme was further applied to overcome the shortcomings of natural enzyme in the application of radiotherapy and would enhance the therapeutic efficacy of radio-enzymatic therapy both in vitro and in vivo.

This manuscript presents new presented intriguing results based on single-atom Cu nanozyme. Of note, this work not only provides an important protocol to design single-atom nanozymes with high catalytic activity and excellent radio-resistance for various applications, for example, radiotherapy, but also offers an alternative guide to other fields including materials science, nanotechnology, and so on. Overall, I strongly recommend its publication on Nature Communications after addressing the following comments.

1. Keep the abbreviations in consistent, for example, "Fourier transformed EXAFS" in page 19 and "FT-EXAFS" in page 6.
2. Additional measurements should be provided to further demonstrate the structural stability of CuN₃-SAzyme after X-ray irradiation, such as atomic-level HAADF-STEM.
3. How about XPS spectra of CuN₃-SAzyme after X-ray irradiation and testing its enzymatic activity?
4. The authors could conduct an in-depth study on the cellular uptake of CuN₃-SAzyme.
5. Does CuN₃-SAzyme have comparable killing effects on other types of tumor cells, such as osteosarcoma K7M2 cells?

6. As demonstrated, CuN₃-SAzyme showed significant cytotoxicity to 4T1 cells. However, what about the cytotoxicity of CuN₄-SAzyme and CuO_x nanozymes?
7. As previously reported (Advanced Materials. 2022, 2205324), nanozymes exhibited catalase-like activity by catalyzing H₂O₂ to generate O₂, which could relieve hypoxia and enhance the effect of radiotherapy. How about the catalase-like activity of CuN₃-SAzyme, dose this activity affect the therapeutic efficacy?
8. Besides long-term toxicity, short-term toxicity of would be also assessed.

Reviewer #2:

Remarks to the Author:

The manuscript entitled “Enhancing Radiation-resistance and Peroxidase-like Activity of Single-atom Copper Nanozyme via Local Coordination Manipulation” theoretically and experimentally demonstrated that a CuN₃-SAzyme shows excellent catalytic performance and also irradiation resistance. Furthermore, CuN₃-SAzyme performs great effect on eradicating cancerous cells without seriously damage to normal tissues. The manuscript is technically sound and the logic is fine. However, I have some questions and suggestions to authors before I can give a judgement on this manuscript.

1. “Synthesis and Characterization of CuN_x-SAzymes” section. I don’t think XPS can provide the element distribution. I don't think this characterization is necessary.
2. In Fig.1 e-g. How the authors confirm the bright dots are Cu atoms but not other contaminations? Is there any possible to give the strong evidence to prove they are single Cu atoms through TEM?
3. “Radiation-resistance characterization of CuN₃-SAzyme.” Section. The authors should provide clear background and knowledge to readers that why this catalytic needs to be radiation resistance under X-ray and NIR. What about other radiation source, such as a α ? γ ?
4. The most drawback of this manuscript is the mechanism of radiation-resistance of this catalytic is not clear. Why this structure of catalytic has strong effect to radiation-resistance? The mechanisms of the resistance need to be studied in depth.

Response to Reviewers' Comments

Reviewer #1 (Remarks to the Author):

The authors designed and synthesized an efficient single-atom nanozyme that consists of densely isolated CuN_3 sites, but not CuN_4 sites by using KCl crystal as the template, for catalytic decomposition of H_2O_2 to yield hydroxyl radicals. Via regulating local coordination number, the resultant CuN_3 -SAzyme exhibited higher peroxidase-like activity than CuN_4 -SAzyme as well as CuO_x nanozymes. This was fully demonstrated by density functional theory calculation. Moreover, this carbon-based nanosheets could stabilize the geometrical structure of CuN_3 sites and maintain the enzyme activity against X-rays. The changes in structure and morphology of CuN_3 -SAzyme after repeated irradiation by X-rays were thoroughly investigated using HAADF-STEM and XAFS measurements. Based on these advantages, this CuN_3 -SAzyme was further applied to overcome the shortcomings of natural enzyme in the application of radiotherapy and would enhance the therapeutic efficacy of radio-enzymatic therapy both in vitro and in vivo.

This manuscript presents new presented intriguing results based on single-atom Cu nanozyme. Of note, this work not only provides an important protocol to design single-atom nanozymes with high catalytic activity and excellent radio-resistance for various applications, for example, radiotherapy, but also offers an alternative guide to other fields including materials science, nanotechnology, and so on. Overall, I strongly recommend its publication on Nature Communications after addressing the following comments.

Response: We truly appreciate the reviewer's encouraging comments. All of the questions raised by this reviewer have been thought and answered elaborately, which greatly improves the quality of our work. Please find our inline responses below.

1. Keep the abbreviations in consistent, for example, "Fourier transformed EXAFS" in page 19 and "FT-EXAFS" in page 6.

Response: Thank you for this question. In the revised manuscript, we have checked carefully and kept all of the abbreviations in consistent, including "Fourier transformed EXAFS" and "FT-EXAFS" as mentioned above.

2. Additional measurements should be provided to further demonstrate the structural stability of CuN_3 -SAzyme after X-ray irradiation, such as atomic-level HAADF-STEM.

Response: We thank the reviewer for bringing up this critical comment that we find to be utmost importance. According to this suggestion, the atomic-level HAADF-STEM image of CuN₃-SAzyme irradiated by X-rays has been added in the revised Supplementary Information (Supplementary Fig. 29c-e). Compared to pre-irradiated CuN₃-SAzyme, bright dots corresponding to the isolated Cu atoms are identified to be dispersed on N-doped carbon nanosheets, as well. In the high magnified TEM image, no obvious Cu-based nanoparticles or clusters are found. These further indicate the high stability of CuN₃-SAzyme against X-rays.

Supplementary Figure 29. Characterization of CuN₃-SAzyme after irradiation by X-rays with the dose of 100 Gy. a HR-TEM image. **b** HAADF-STEM image and corresponding EDS mapping. **c** Atomic-level HAADF-STEM image. **d** Enlarged HAADF-STEM image of the marked area in **c**. **e** Corresponding surface intensity map of **c**, the yellow dots are Cu atoms. Three times each morphology characterization was repeated independently with similar results. Representative images are presented.

Also, we perform additional density functional theory (DFT) calculations and extensive *ab initio* molecular dynamics (AIMD) simulations to fully understand the structural stability of CuN₃-SAzyme under X-ray irradiation. For further details, please see the response to **Reviewer #2 (Comment 4)**. In addition, we investigate the excellent radioresistance of CuN₃-SAzyme against γ -rays which have higher energy (a cobalt-60 source, average photon energy of 1.25 MeV) (Supplementary Figs. 37-39). Collectively, all of these results clearly demonstrate the good structural stability of CuN₃-SAzyme under both X-ray and γ -ray irradiation.

Supplementary Figure 37. Evaluation of the specific activities of CuN₃-SAzyme after irradiation by γ -rays with the radiation dose of 20 Gy/100 Gy (pH 3.54). These data are presented as mean values \pm SD (n = 3 independent experiments).

Supplementary Figure 38. Characterization of CuN₃-SAzyme after irradiation by γ -rays with the radiation dose of 100 Gy (a-c) and 500 Gy (d-f). a,d Atomic-level HAADF-STEM image. b,e Enlarged HAADF-STEM image of the marked area in a (b) or d (e). c,f Corresponding surface intensity map of a (c) or d (f). The yellow dots are Cu atoms. Three times each morphology characterization was repeated independently with similar results. Representative images are presented.

Supplementary Figure 39. Comparison of XAFS spectra of CuN₃-SAzyme before/after γ -ray irradiation (500 Gy). a Cu K-edge XANES spectra. **b** Fourier-transformed magnitudes of experimental Cu K-edge EXAFS signals at R space. **c** k -space plots. **d** q -space plots.

3. How about XPS spectra of CuN₃-SAzyme after X-ray irradiation and testing its enzymatic activity?

Response: Thank you for your valuable suggestion which would help to further understand the changes in the structure and enzymatic activity of CuN₃-SAzyme during X-ray irradiation. Accordingly, we have added these additional data in the revised Supplementary Information (Supplementary Fig. 28). The results of XPS analysis demonstrate the presence of C, N, and Cu elements in the irradiated CuN₃-SAzyme, and no significant changes in its C 1s and N 1s spectra. For Cu 2p spectrum, irradiated CuN₃-SAzyme still contains Cu in the states of Cu⁺ and Cu²⁺, and the amount of Cu²⁺ increases to a certain extent after irradiation with X-rays. In combination to the result of XAFS analysis, it is inferred that there is no obvious change in the coordination environment of CuN₃-SAzyme during irradiation by X-rays, indicating its good structural stability. The change in the chemical state of Cu element in CuN₃-SAzyme may be due to the interaction between Cu atom with X-rays.

Supplementary Figure 28. XPS analysis of CuN₃-SAzyme after X-ray irradiation. a XPS survey spectrum. **b-d** Cu 2p (**b**), N 1s (**c**), and O 1s (**d**) XPS spectra.

For the enzymatic activity, the apparent specific activity of irradiated CuN₃-SAzyme is calculated to be 11.33 U mg⁻¹ which is comparable to that of pre-irradiated CuN₃-SAzyme (Fig. 3c). Ever the radiation dose reaches up to 100 Gy, there is only slight decrease in its specific activity (10.79 U mg⁻¹). Besides, we also determined the enzymatic activity of CuN₃-SAzyme after γ -ray irradiation. As expected, there are no changes in its enzymatic activity: the apparent specific activity is calculated to be 12.28 U mg⁻¹ for 20 Gy and 13.01 U mg⁻¹ for 100 Gy, respectively (Supplementary Fig. 28). These further demonstrate that CuN₃-SAzyme has good structural stability against X-rays and γ -rays and thus maintains its enzymatic activity beyond natural enzymes.

4. The authors could conduct an in-depth study on the cellular uptake of CuN₃-SAzyme.

Response: Thank you for this valuable advice because understanding uptake of nanomaterials is fundamentally important to determine the successful delivery of CuN₃-SAzyme into cancer cells. Therefore, we further studied intercellular uptake routes of CuN₃-SAzyme by inhibiting specific endocytic pathways (*Nat Nanotechnol* **2021**, *16*, 266). After incubating 4T1 cells with CuN₃-SAzyme and inhibitors

(including cytochalasin D and wortmannin) (*Acta Pharm Sin B* **2021**, *11*, 903), there is no significant decrease in cellular uptake (Supplementary Fig. 44). However, incubation of 4T1 cells at 4 °C leads to obviously reduced cellular uptake (from 63.9% to 50.7%). Moreover, the inhibitors of filipin and chlorpromazine show more significant effect on the uptake of CuN₃-SAzyme, decreasing to 45.1% for filipin and to 35.2% for chlorpromazine. These indicate that the uptake of CuN₃-SAzyme occurs not by passive diffusion or micropinocytosis but likely by clathrin-mediated endocytosis.

Supplementary Figure 44. Flow cytometry histograms illustrating cellular uptake and inhibition of uptake of CuN₃-SAzyme.

5. Does CuN₃-SAzyme have comparable killing effects on other types of tumor cells, such as osteosarcoma K7M2 cells?

Response: Thank you for this kind suggestion. According to this comment, we further investigated the cell viability of K7M2 cells after incubation with CuN₃-SAzyme (Supplementary Fig. 45). Similar to 4T1 cells, the proliferation is clearly reduced by CuN₃-SAzyme in a dose-dependent manner after 24 h of incubation, with the IC₅₀ of 18.50 μg mL⁻¹, confirming that CuN₃-SAzyme also enables the generation of high-toxic ·OH radicals through peroxidase-like catalytic reaction to kill cancer cells.

Supplementary Figure 45. Viabilities of K7M2 cells after incubation with CuN_3 -SAzyme. These data are presented as mean values \pm SD ($n = 6$ independent experiments).

6. As demonstrated, CuN_3 -SAzyme showed significant cytotoxicity to 4T1 cells. However, what about the cytotoxicity of CuN_4 -SAzyme and CuO_x nanozymes?

Response: Thank you for pointing this out. The cytotoxicity of CuN_4 -SAzyme, Cu_2O , and CuO nanozymes was evaluated by CCK-8 kit. As shown in Supplementary Fig. 46, both CuN_4 -SAzyme and CuO_x nanozyme exhibits less toxicity at the same incubation concentration (in terms of Cu element). These results are consistent with the enzymatic assay and confirm that CuN_3 -SAzyme enables the generation of more $\cdot\text{OH}$ radicals than CuN_4 -SAzyme, Cu_xO nanozymes within cancer cells, leading to higher killing efficacy.

Supplementary Figure 46. Viabilities of 4T1 cells after incubation with CuO nanozyme (a), Cu_2O nanozyme (b), and CuN_4 -SAzyme (c). These data are presented as mean values \pm SD ($n = 6$ independent experiments).

7. As previously reported (Advanced Materials. 2022, 2205324), nanozymes exhibited catalase-like activity by catalyzing H_2O_2 to generate O_2 , which could relieve hypoxia and enhance the effect of radiotherapy. How about the catalase-like activity of CuN_3 -SAzyme, dose this activity affect the therapeutic efficacy?

Response: Thank you for this kind advice. The catalase-like activity of CuN_3 -SAzyme was further determined and calculated to be 0.40 U mg^{-1} (Supplementary Fig. 14). This value is much lower than that for its peroxidase-like activity (11.33 U mg^{-1}) as well as that for the catalase-like activity of FeN_4 -SAzyme (52.64 U mg^{-1}) as previously reported (*Adv Mater* **2022**, 34, 2205324). Given that the catalase-like catalytic reaction competes with the peroxidase-like reaction, it is believed that CuN_3 -SAzyme mainly catalyze the oxidation of H_2O_2 to yield $\cdot\text{OH}$ radicals. Therefore, the catalase-like activity of CuN_3 -SAzyme has limited influence on its therapeutic efficacy.

Supplementary Figure 14. The catalase-like activity of CuN_3 -SAzyme. These data are presented as mean values \pm SD ($n = 3$ independent experiments).

8. Besides long-term toxicity, short-term toxicity of would be also assessed.

Response: Thank you for this valuable advice. To evaluate the short-term toxicity of CuN_3 -SAzyme, the biodistribution, blood routine, blood biochemical, and body weight of mice subcutaneously injected by CuN_3 -SAzyme were investigated. Between the control group and treated groups, there is no significant differences in body weight (Supplementary Fig. 61) and the levels of copper element in the heart, liver, spleen, lung, and kidney (Supplementary Fig. 66). Remarkably, injected CuN_3 -SAzyme mainly stays at the injection site and does not move to major organs. Moreover, the value of blood routine test indexes and blood biochemistry test indicators fall within

the reference ranges of the control (Supplementary Fig. 65). In addition, no apparent histological changes including inflammation and organ damage are observed in mice treated with CuN₃-SAzyme at 10 and 20 mg kg⁻¹ doses (Supplementary Fig. 63). All of these results confirm the good biocompatibility, therefore CuN₃-SAzyme can be used as an effective yet safe therapeutic agent for cancer-specific treatment.

Supplementary Figure 61. Body weight of mice subcutaneously injected by CuN₃-SAzyme. All data are presented as means \pm SD (n = 6 biologically independent animals).

Supplementary Figure 63. Representative H&E images of main organs including heart, liver, spleen, lung, and kidney of mice subcutaneously injected by CuN₃-SAzyme. Experiments were performed three times with similar results. Representative images are presented. Scale bar: 50 μ m.

Supplementary Figure 65. Blood routine test indexes and blood biochemistry test indicators of mice subcutaneously injected by CuN₃-SAzyme. All data are presented as means \pm SD (n = 6 biologically independent animals).

Supplementary Figure 66. Biodistribution of CuN₃-SAzyme subcutaneously injected. All data are presented as means ± SD (n = 6 biologically independent animals).

Reviewer #2 (Remarks to the Author)

The manuscript entitled “Enhancing Radiation-resistance and Peroxidase-like Activity of Single-atom Copper Nanozyme via Local Coordination Manipulation” theoretically and experimentally demonstrated that a CuN₃-SAzyme shows excellent catalytic performance and also irradiation resistance. Furthermore, CuN₃-SAzyme performs great effect on eradicating cancerous cells without seriously damage to normal tissues. The manuscript is technically sound and the logic is fine. However, I have some questions and suggestions to authors before I can give a judgement on this manuscript.

Response: Thanks for the reviewer’s positive feedback, constructive comments, and very valuable suggestions, which help us a lot to improve the quality of the manuscript. Below please find our point-by-point responses.

1. “Synthesis and Characterization of CuN_x-SAzymes” section. I don't think XPS can provide the element distribution. I don't think this characterization is necessary.

Response: Thank you for this valuable suggestion. First, we truly appreciate the reviewer’s attentive review and apologize for failing to explain clearly enough in the manuscript. As mentioned by this reviewer, the result of XPS analysis cannot provide the information on the element distribution. Indeed, the description “*This is further demonstrated by X-ray photoelectron spectroscopy (XPS) analysis*” wants to explain that the result of XPS analysis could demonstrate the presence of C, N, and Cu elements throughout the whole architectures of CuN₃-SAzyme and CuN₄-SAzyme, respectively. To avoid misunderstanding by readers, we have changed this description in the revised manuscript, and XPS analysis is mainly applied to determine the chemical states of elements on the surface of CuN₃-SAzyme and CuN₄-SAzyme.

2. In Fig.1 e-g. How the authors confirm the bright dots are Cu atoms but not other contaminations? Is there any possible to give the strong evidence to prove they are single Cu atoms through TEM?

Response: First, we truly thank the reviewer for this critical comment. As known, the HAADF-STEM image is a *Z* (i.e., the atomic number) contrast image, in which element with higher *Z* exhibits stronger contrast in the dark-field TEM image, thereby appearing as bright spots. Because the atomic number of Cu atoms (*Z*=29) is much higher than those of C (*Z*=6) and N (*Z*=7) atoms, these bright dots are largely possible to be Cu atom. To further prevent interference with other contaminations, we added the atomic-level electron energy-loss spectrum. As shown in Supplementary Fig. 8, a distinctive Cu signal is detected in the EELS spectrum of CuN₃-SAzyme. These strongly demonstrate that the bright dots in the dotted box indeed are single Cu atoms.

Supplementary Figure 8. HAADF-STEM image of CuN₃-SAzyme (a) and corresponding EELS spectrum of Cu element from the dashed box (b). Three times each morphology characterization was repeated independently with similar results. Representative images are presented.

3. “Radiation-resistance characterization of CuN₃-SAzyme.” Section. The authors should provide clear background and knowledge to readers that why this catalytic needs to be radiation resistance under X-ray and NIR. What about other radiation source, such as a α ? γ ?

Response: We thank the reviewer for the insightful comment. As demonstrated by our results, the peroxidase-like enzymatic activity of CuN₃-SAzyme can be remarkably enhanced under 808 nm NIR light illumination and X-ray irradiation. This external field-enhanced peroxidase-like enzymatic activity facilitates the increase the deposition of radiation energy via generating more \cdot OH radicals from H₂O₂, therefore enabling it as a potential radiosensitizer to enhance the therapeutic efficacy of radiotherapy. On the other hand, for clinic radiotherapy treatment, the total radiation dose for patients may be required up to 70 Gy (including X-rays and γ -rays). Therefore, as a radiosensitizer, it is required that CuN₃-SAzyme has excellent radiation-resistance against X-/ γ -rays as well as NIR light, that is, CuN₃-SAzyme should maintain its original chemical structure and peroxidase-like activity under irradiation, so that robust radiosensitization effect would be continuously achieved to enhance therapeutic efficacy during the whole therapeutic period. This is why CuN₃-SAzyme needs to be radiation-resistant under 808 nm NIR light and X-ray irradiation. In the revised manuscript, we have followed this valuable suggestion and accordingly provided a detailed description. We hope that the additional description may help the readers to understand this research paper in depth.

According to this comment, we also further investigate the radioresistance of CuN₃-SAzyme against γ -rays which have higher energy (a cobalt-60 source, average photon energy of 1.25 MeV) (Supplementary Figs. 37-39). Please see the response to **Reviewer #1 (Comment 2)**. The results suggest that there are no significant changes in the enzymatic activity of CuN₃-SAzyme or its chemical structures after γ -ray

irradiation. Collectively, all of these results clearly demonstrate the good structural stability of CuN₃-SAzyme under both X-ray and γ -ray irradiation.

4.: The most drawback of this manuscript is the mechanism of radiation-resistance of this catalytic is not clear. Why this structure of catalytic has strong effect to radiation-resistance? The mechanisms of the resistance need to be studied in depth.

Response: We thank the reviewer for the insightful comments, which inspire us to think more about the mechanisms for the radiation-resistance of CuN₃-SAzyme. To answer these questions, we have performed additional density functional theory (DFT) calculations and extensive *ab initio* molecular dynamics (AIMD) simulations, where the simulation model of the latter contains a total of 450 atoms (Supplementary Fig. 33).

In the irradiation, X-rays can excite electrons in the inner layers of constituent atoms, causing not only radiative but also non-radiative transitions. The non-radiative transitions could induce a drastic energetic stimulation and lead to further ionization of the matter. Considering that it is extremely complex to directly simulate the structural evolution of the matter under X-ray irradiation from first principles, here, alternatively, we focus on whether CuN₃-SAzyme can maintain its structural stability under drastic energy fluctuations (equivalent to being in a hot environment of thousands of Kelvin) and when one to several electrons are ionized, both of which can be considered as the potential influences of X-ray irradiation. It is found that the CuN₃ active sites of CuN₃-SAzyme are well preserved at an equivalent temperature as high as 1,773 K (Supplementary Fig. 34), demonstrating that such CuN₃ active sites are indeed very stable. Only a small fraction of Cu atoms could detach from the anchoring N₃ groups when the temperature is further increased to 2,273 K, corresponding to more drastic energy fluctuations, (Supplementary Fig. 35), consistent with the observed slight decrease in the peroxidase-like activity of CuN₃-SAzyme under high-dose X-ray irradiation (500 Gy). Moreover, the CuN₃ active sites remain stable even when the simulation model ionizes one, two, three, and four electrons under the same energetic stimulation equivalent to 1,773 K (Supplementary Fig. 36). Our results reproduce the experimentally observed excellent stability of CuN₃-SAzyme and further reveal the mechanisms for the corresponding irradiation-resistance, as follows:

The first and perhaps the most important reason is that each CuN₃ active site of CuN₃-SAzyme contains only one Cu atom and three N atoms, and the interaction between them is sufficiently strong. This means that the CuN₃ active sites can maintain not only structural but also conformational stability. Our DFT calculations show that the interaction strength between the Cu atom and the N₃ group is as high as 2.96 eV, indicating that the anchored Cu atom is not easily detached from the substrate. Moreover, our AIMD simulations at 1,773 K for simulating the energy fluctuations of CuN₃-SAzyme under X-ray irradiation have not found any other geometry of CuN₃ except the one shown in Fig. 2f, demonstrating that the CuN₃

active sites can be well preserved and are not susceptible to conformational changes (Supplementary Fig. 34). When one, two, three, and four electrons are removed from CuN₃-SAzyme, to simulate the ionization of CuN₃-SAzyme under X-ray irradiation, the above results still hold (Supplementary Fig. S36), further confirming the stability of the CuN₃ active sites. In contrast, the active center of the natural enzyme (HRP) used in the experiments contains many more atoms and groups, and the interactions include not only strong chemical bonds but also weaker hydrogen bonds and intermolecular interactions. Under X-ray irradiation, these weaker interactions can be easily disrupted, causing the active center to change to other configurations and lose its original enzymatic activity. Such instabilities can be further aggravated by electron ionization caused by X-ray irradiation. The above differences between CuN₃-SAzyme and natural enzyme make the former much more resistant to X-rays.

The second reason stems from the low Cu content (2.98 wt%) in CuN₃-SAzyme, which means that the short-wavelength X-ray photons that penetrate the nanozyme have a high probability of interacting with the carbon architecture and a very low probability of interacting with the CuN₃ active sites directly. Previous experiments have shown that the main structure of few-layer graphene remains unchanged after X-ray irradiation (*Nucl. Instrum Methods Phys Res B* **2013**, 304, 49), and our AIMD simulation results also confirm the stability of CuN₃-SAzyme (Supplementary Figs. 34,36). In contrast, the active center of the natural HRP enzyme occupies a much larger space than that of the CuN₃ moieties, and the stability of the adopted conformation of the active center also depends on its surrounding environment with an even larger size. The X-ray photons can easily hit some of the components associated with the functionality of the natural enzyme, thus destabilizing the conformation and decreasing its enzymatic activity. This is another reason why CuN₃-SAzyme shows better radiation-resistance against X-rays than the natural enzyme.

Supplementary Figure 33. The simulation model used for the AIMD simulations, containing a total of 414 carbon, 27 nitrogen, and 9 copper atoms. Gray: C; Blue: N; Light coral: Cu. The boundary of the supercell is shown in pinkish brown.

Supplementary Figure 34. Time evolution of the distribution function $g(r)$ for the distance between Cu and N atoms in different time periods during the AIMD simulations at 1,773 K. It is obvious that the Cu-N bonds can be well maintained.

Supplementary Figure 35. Snapshots of the CuN_3 -SAzyme model at different moments in the AIMD simulations at 2,273 K. One of the Cu atoms has detached from the N_3 group in the second half of the simulation (red circles). The disruption of the CuN_3 active moiety becomes more pronounced as the temperature is further increased to 2,773 K and 3,273 K.

Supplementary Figure 36. Time evolution of the distribution function $g(r)$ for the distance between Cu and N atoms in different time periods during the AIMD simulations at 1,773 K after one (a), two (b), three (c), and four (d) electrons have been ionized. The ionization occurs after the AIMD simulations at 1,773 K have run for 5 ps in Supplementary Fig. S34. It is obvious that Cu-N bonds are still well maintained.

Reviewers' Comments:

Reviewer #1:

Remarks to the Author:

The authors have addressed all my concerns and I highly suggest accepting the manuscript.

Reviewer #2:

Remarks to the Author:

I only have one more suggestion. I believe that the bright dots are copper now. But the EELS result is not convincing. I suggest to remove the supplementary fig.8.

Reviewer #3:

Remarks to the Author:

This manuscript named "Enhancing Radiation-resistance and Peroxidase-like Activity of Single-atom Copper Nanozyme via Local Coordination Manipulation" does a good job of describing the mechanisms for the radiation-resistance of CuN₃-SAzyme through DFT and AIMD simulations. There are no comments for the computational part of this work. I recommend that this article be accepted and published in Nature Communications.

Reviewers' Comments:

Reviewer #1 (Remarks to the Author):

The authors have addressed all my concerns and I highly suggest accepting the manuscript.

Reviewer #2 (Remarks to the Author):

I only have one more suggestion. I believe that the bright dots are copper now. But the EELS result is not convincing. I suggest to remove the supplementary fig.8.

Reviewer #3 (Remarks to the Author):

This manuscript named "Enhancing Radiation-resistance and Peroxidase-like Activity of Single-atom Copper Nanozyme via Local Coordination Manipulation" does a good job of describing the mechanisms for the radiation-resistance of CuN₃-SAzyme through DFT and AIMD simulations. There are no comments for the computational part of this work. I recommend that this article be accepted and published in Nature Communications.

Response to Reviewers' Comments

Reviewer #1 (Remarks to the Author):

Comment: The authors have addressed all my concerns and I highly suggest accepting the manuscript.

Response: We truly appreciate the reviewer's careful review and constructive suggestions that have helped us improve the clarity and accuracy of our manuscript.

Reviewer #2 (Remarks to the Author):

Comment: I only have one more suggestion. I believe that the bright dots are copper now. But the EELS result is not convincing. I suggest to remove the supplementary fig.8.

Response: We really appreciate the reviewer's positive comments. According to this advice, Supplementary Fig. 8 has been removed in the revised manuscript and revised Supplementary Information.

Reviewer #3 (Remarks to the Author):

Comment: This manuscript named "Enhancing Radiation-resistance and Peroxidase-like Activity of Single-atom Copper Nanozyme via Local Coordination Manipulation" does a good job of describing the mechanisms for the radiation-resistance of CuN₃-SAzyme through DFT and AIMD simulations. There are no comments for the computational part of this work. I recommend that this article be accepted and published in Nature Communications.

Response: We really appreciate the reviewer's positive comments.